# Effects of Different Types of Exercise Training on Pulmonary Arterial Hypertension: A Systematic Review

**DOI:** 10.3390/jcm9061689

**Published:** 2020-06-02

**Authors:** Lena Waller, Karsten Krüger, Kerstin Conrad, Astrid Weiss, Katharina Alack

**Affiliations:** 1Department of Exercise Physiology and Sports Therapy, Institute of Sports Sciences, Justus-Liebig-University Giessen, 35394 Giessen, Germany; karsten.krueger@sport.uni-giessen.de (K.K.); kerstin.conrad@sport.uni-giessen.de (K.C.); katharina.alack@sport.uni-giessen.de (K.A.); 2Department of Internal Medicine, Institute of Pulmonary Pharmacotherapy, Justus-Liebig-University Giessen, Universities of Giessen and Marburg Lung Center (UGMLC), 35392 Giessen, Germany; astrid.weiss@innere.med.uni-giessen.de

**Keywords:** pulmonary arterial hypertension, exercise training, human studies, experimental models

## Abstract

Pulmonary arterial hypertension (PAH) represents a chronic progressive disease characterized by high blood pressure in the pulmonary arteries leading to right heart failure. The disease has been a focus of medical research for many years due to its worse prognosis and limited treatment options. The aim of this study was to systematically assess the effects of different types of exercise interventions on PAH. Electronic databases were searched until July 2019. MEDLINE database was used as the predominant source for this paper. Studies with regards to chronic physical activity in adult PAH patients are compared on retrieving evidence on cellular, physiological, and psychological alterations in the PAH setting. Twenty human studies and 12 rat trials were identified. Amongst all studies, a total of 628 human subjects and 614 rats were examined. Regular physical activity affects the production of nitric oxygen and attenuates right ventricular hypertrophy. A combination of aerobic, anaerobic, and respiratory muscle training induces the strongest improvement in functional capacity indicated by an increase of 6 MWD and VO_2_ peak. In human studies, an increase of quality of life was found. Exercise training has an overall positive effect on the physiological and psychological components of PAH. Consequently, PAH patients should be encouraged to take part in regular exercise training programs.

## 1. Introduction

The following systematic review addresses the effects of different types of exercise training on molecular, physiological, and psychological alterations in pulmonary arterial hypertension (PAH), which is a devastating and life-threatening disease of the lung vasculature.

### 1.1. Rationale

PAH has been a focus of medical research for many years and is an important issue due to its worse prognosis and limited treatment options. It is a progressive deteriorating disease which remains incurable with a median survival of 5–7 years [1] and an incidence of 2–7.6 newly diagnosed cases per one million per year (depending on the different international registries) [2]. Galiè et al. [3] and Hoeper [4] found a prevalence of 15–60 cases per one million adults in Europe, most commonly diagnosed with idiopathic PAH (iPAH), and demonstrated women as predominantly affected compared to men. PAH is characterized by an increased pulmonary vascular resistance (PVR) due to an augmented vascular remodeling through inflammation, proliferation, and vasoconstriction and progressively leads to right ventricular hypertrophy (RVH), further resulting in right heart failure and premature death [5]. The disease is hemodynamically defined by a mean pulmonary arterial pressure (mPAP) ≥ 20 mmHg at rest, with the presence of a pulmonary arterial wedge pressure (PAWP) of ≤ 15 mmHg and a pulmonary vascular resistance (PVR) > 3 Wood units (WU) [6]. As PAH develops, pathological alterations impact on both physiological and psychological factors in patients with various symptoms occurring, the most notable being shortness of breath, fatigue, and a decreasing quality of life (QoL), as the severity of the disease advances [7,8]. Therefore, it is of great medical importance to assess effective treatment interventions in order to mitigate and potentially reverse PAH-related symptoms. Exercise training (ET) was considered inappropriate for many years, as it was hypothesized it could have a negative impact on the cardiovascular system and result in premature death [9]. Nevertheless, the beneficial effect of ET on various chronic diseases was proven, thus a shift in disease management and therapeutic options for PAH became of interest [10]. Mereles et al. [11] was the first to examine the favorable effects of ET on severe forms of PAH through a randomized controlled trial (RCT), followed by further studies investigating safety and efficacy of ET, reporting an improvement particularly in QoL and exercise capacity [12]. These scientific advances led to updated recommendations of a combinatory pharmaceutical and exercise orientated treatment in PAH [12,13,14]. Additionally, recent research has shown that health care costs for patients with combined drug and exercise therapy are lower than for those solely treated with medication [15].

Despite this constant scientific progress, therapeutic effects of various types of physical activities on the complex pathology and pathological alterations in PAH are not completely understood [16]. Hence, it continues to be of great interest to investigate different forms of exercise interventions and to develop a deeper understanding of its effectiveness on molecular, physiological, and psychological factors.

### 1.2. Objectives

This systematic review aims to assess and summarize the effects of different types of regular aerobic and anaerobic exercise interventions on PAH. Studies with regards to chronic physical activity in adult PAH patients are compared with specific emphasis on retrieving evidence on molecular, physical, and psychological alterations in the PAH setting. The current state of scientific knowledge on PAH is systematically analyzed in order to propose future research approaches and to develop a better understanding of therapeutic impacts of diverse ET.

## 2. Pulmonary Arterial Hypertension

### 2.1. Classification of Pulmonary Hypertension

Pulmonary hypertension (PH) is characterized by high blood pressure within the pulmonary circulation vasculature. This occurs due to different etiology and was previously categorized into a simple primary and secondary form. However, a specified classification was established in 1998 presenting new scientific aspects of similarities in pathology, hemodynamic characteristics and management [17]. In 2018, during the 6th World Symposium on Pulmonary Hypertension the classification of pulmonary hypertension was further refined and now comprises of five groups: Group 1—PAH; Group 2—PH due to left heart disease; Group 3—PH due to lung diseases and/or hypoxia; Group 4—PH due to pulmonary artery obstructions; and Group 5—PH with unclear and/or multifactorial mechanisms [6]. Group 1 PH (PAH) is identified through different causes and divided into further subcategories, as shown in Table 1.

In PAH and across all other forms of PH, the precapillary pressure in lung vessels is increased and characterized by a mPAP ≥ 20 mmHg at rest as measured during right heart catheterization (RHC), which is the standard diagnostic tool. Because this single value of mPAP can neither fully characterize the clinical condition nor describe the pathological process per se, PAWP as well as cardiac output (CO) and pulmonary vascular resistance (PVR = (mPAP–PAWP)/CO) were proposed to be taken into account, as stated in an updated version of the hemodynamic definitions of PH according to the Sixth World Symposium on Pulmonary Hypertension (WSPH) in 2018. Now, pre-capillary PH, i.e., PAH, is defined by mPAP > 20 mmHg with PAWP ≤ 15 mmHg and PVR ≥ 3 WU, as determined by right heart catheterization. Thereby, PAH is clearly distinguished from isolated post-capillary PH (IpcPH) by mPAP > 20 mmHg with PAWP > 15 mmHg and PVR < 3 WU. Combined pre- and post-capillary PH (CpcPH) is described by mPAP > 20 mmHg and PAWP > 15 mmHg and PVR ≥ 3 WU [6]. In previous definitions, a mean pressure in the pulmonary arteries ≥ 30 mmHg during exercise was an additional criterion to make a reasonable diagnosis. However, this was removed due to the findings that the mean PAP varies in a wide range depending on the type of exercise, the patients training status, and age [4]. Furthermore, Montani et al. [18] explained missing evidence that adopting a definition on exercise as the mean PAP also shows a large variability in healthy individuals. However, exercise-induced PH is often considered as an early, intermediary phase of pre-capillary PH and not only this particular silent pathology is commonly identified by physical exercise tests during medical diagnosis [19]. Although not listed as a sub-group of PH, there is emerging consensus to define exercise-induced PH by a mPAP > 30 mmHg at a CO < 10 L/min and a total PVR > 3 WU at maximum exercise, in the absence of PH at rest [20]. While having normal clinical parameters at rest, those patients present with the same symptomatic (e.g., dyspnea on exertion), which is associated with decreased exercise capacity and poor prognosis with yet hidden heart failure and/or pulmonary vascular disease as underlying causes. As the right ventricular afterload and the RV capacity increase during the progression of the disease or under physical activity, any preexisting condition with impaired RV function enhances the risk of RV failure. Therefore, this type of PH is often indicative and of prognostic relevance in mitral valve disease, aortic stenosis, heart failure, systemic sclerosis, chronic obstructive pulmonary disease, and symptomatic patients after pulmonary endarterectomy [19]. As the healthy right ventricle (RV) is susceptible to injury if exercise is sufficiently extreme leading to chronic RV damage and arrhythmias [21], it is overall well accepted that a pathologic right ventricle may be even more sensitive to exercise-induced injury and the related PH symptomatic. For further insights into this interesting topic which is not in the main focus of this systematic review due to limitations, the reader is referred to excellent reviews from experts on this field.

### 2.2. Pathogeneses and Clinical Manifestation

The pathogenesis of PAH underlays a very complex chain of mutually influencing molecular, cellular, and physiological factors, and, despite constant scientific advances, it is still not completely understood [5,22]. Generally speaking, the cardiopulmonary and cardiovascular system plays an essential role in vital physiological processes. The primary function of the pulmonary and systemic circulation is to transport oxygen (O_2_) and nutrients into the cells and simultaneously metabolic products such as carbon dioxide (CO_2_) out of the body. The gas exchange of O_2_ and CO_2_ occurs in the small alveoli and surrounding capillaries of the lungs. The two gases then diffuse through thin cell layers between alveoli and pulmonary capillaries and O_2_ binds to erythrocytes allowing for CO_2_ to be exhaled [23]. In patients with PAH, the perfusion of pulmonary vessels is impaired due to obstructive remodeling and narrowing caused by vascular changes in the walls of the pulmonary arteries forming so-called plexiform lesions, which are comprised of a plexus of capillary-like channels. Additionally, unavoidable hemodynamic changes are characterized by a reduction in pulmonary blood vessels which hinders normal blood flow [5]. Subsequently, PVR and precapillary blood pressure increases, causing further pathological alterations and clinical symptoms in affected patients [24].

The latest research on underlying pathogenic mechanisms, which cause pulmonary arterial obstruction and right ventricular (RV) failure leading to premature death, is not completely understood but still ongoing [5]. However, scientific progress has already shown that the narrowing in the pulmonary vasculature seems to occur as a result of dysfunctions in cellular and molecular processes and signaling pathways [16]. These multifactorial pathways are comprised of a wide range of interacting cellular and molecular factors resulting in pathological intimal hyperplasia, medial thickening, and adventitial remodeling in the lungs [18]. The proliferation of smooth muscle cells (SMC) and endothelia cells (EC) in the tunica intima and media is increased in PAH patients, as is the remodeling of the pulmonary vasculature. Further endothelial dysfunctions, such as abnormal potassium functions or a lack of vasodilators such as nitric oxide (NO) can lead to an increased vasoconstriction [18]. Only three of the pathogenic pathways driving the development of PAH are sufficiently well characterized—the prostacyclin, the endothelin, and the NO pathways [16]. In addition, augmented inflammatory mediators and thrombosis also seem to play an important role in the disease progression [25]. Moreover, the development of PAH may be impacted by genetic predispositions and environmental risk factors [18,26].

In the further course of pulmonary vascular obstruction and impaired hemodynamics, mean PAP and PVR increases causing RV afterload to also increase [26]. The right heart compensates for the increased afterload and PAP through right ventricular hypertrophy (RVH), which is associated with alterations in the cardiac output (CO) [27]. As right ventricular function is the most crucial parameter dictating the patients’ survival, its capability to physiologically adjust to these increased pressure conditions is indispensable. Therefore, RVH is a logical consequence to initially maintain right ventricle (RV) contractility to ensure proper heart stroke volume and cardiac output (“adaptation”). These cellular processes are needed to cope with the elevated afterload and the general demand for oxygen saturation of the pulmonary but also the systemic circulation. This means, in PH, the right ventricle adapts to the increasing vascular afterload by enhancing contractility (“coupling”) to maintain blood flow [28]. During disease progression, dilatation of the RV occurs in an attempt to limit the reduction in stroke volume. Cardiac fibrosis and increased stiffness can be observed which escalates into enhanced wall stress and ultimately into decreased cardiac performance (“uncoupling”). This phase of “mal-adaptation” finally leads to right heart failure associated death [29]. As the mutually destructive changes in the pulmonary and cardiac system progress, the affected patients begin to suffer from various unspecific symptoms [18]. Matura et al. [8] described dyspnea, fatigue, and a decreased functional capacity as the most notable symptoms in PAH. The cardiopulmonary alterations subsequently limit a patient’s exercise capacity and reduce their ability to perform physical activity. Additionally, the symptoms of the disease are known to take a toll on afflicted patients leading to psychological distress and a diminished QoL [7].

The heterogeneous course and progression of the disease and its unspecific symptoms, as shown in Figure 1, make it difficult to diagnose and establish a specific treatment for PAH. Calcium channel blockers inhaled nitric oxide (NO) together with prostacyclin analogs (e.g., epoprostenol) aim to counteract the vasoconstrictive phenotype of pulmonary vascular cells. Prostacyclin analogs administration results in elevated concentrations of cAMP, promoting vasodilation and inhibiting both platelet aggregation and smooth muscle cell proliferation. Selexipag further stimulates intracellular prostacyclin signaling as it binds to the IP receptor which promotes vasodilation and interferes with proliferation. If patients are not vasoreactive, targeted PAH therapies are applied including phosphodiesterase 5 (PDE5) inhibitors such as sildenafil (or tadalafil), which target the NO pathway by preventing the degradation of cGMP. In addition, soluble guanylate cyclase (sGC) stimulators such as riociguat also act on the very same signaling pathway by increasing the enzymatic activity of sGC independent from endogenous NO leading to increased cGMP levels. Finally, the endothelin receptor antagonists (ERAs) bosentan, ambrisentan, and macitentan counterbalance the vasoconstrictive mode of action of endothelin (ET-1), which is a known vasoconstrictor [30]. In addition to this PAH-specific therapy, anticoagulants and diuretics are commonly used in practice. Current scientific advances continue to reveal new approaches in terms of pathogenesis, diagnosis and management [18]. Various diagnostic tools, such as RHC, cardiopulmonary exercise testing (CPET), and blood analysis, are important for an initial categorization of PH patients. The analyzed diagnostic factors have been found to closely link to the severity level of the disease and to prognostic values [31]. The latest research of two notable prognostic factors has shown that a decrease in N-terminal-pro-brain-natriuretic-peptide (NT-proBNP) levels predict a better long-term outcome in PAH patients [32]. Furthermore, these levels correlate with patient’s functional capacity, which is again conversely related to an increase in the 6-min walking distance (6 MWD) and an improvement of the patient’s functional capacity during CPET [32,33]. By determining the patients PAH status and severity through the Benza et al. [31] validated risk score, therapeutic approaches can be derived. From that point, management of PAH has recently been found to be more effective when a combination of drug and exercise therapy is administered [13].

### 2.3. The role of Exercise

In medicine, a disease is examined under etiological, pathogenic, and therapeutic aspects. For the treatment, pharmaceutical therapies are often used primarily to result in an improvement and possibly a cure. However, the aspect of non-pharmaceutical therapy, such as exercise therapy and/or restrictive diets [34], takes an equally important contribution and is of increasing interest in current disease management [10]. Regarding other chronic diseases, physical activity was found beneficial, which was reinforced by Chia et al. [35] in topical research for PAH. The question whether performing ET can be safely and efficaciously included in the treatment of PAH has already been examined in recent studies [12,36,37]. In comparison to sedentary behavior, no risk or associated adverse events could be found while conducting ET complementary to drug therapy; there was even a significant improvement in PAH-related symptoms detected [38]. Despite incompletely traceable mechanisms of the favorable effects of ET on PAH and ongoing scientific research, a combination of drug and exercise therapy is now recommended in PH treatment guidelines [3,18].

Physical activity and ET are long known to have an impact on the human body at a molecular, cellular, physiological, and psychological level, and by taking a closer look at the cardiopulmonary system in the context of PAH, beneficial adaptations can be found. As the cardiopulmonary system holds the two key components contributing to the progression of the disease—obstructive remodeling in pulmonary arteries [39] and right heart dysfunction [40]—it is of interest to look at potential influences of ET on all the coherent mechanisms. This ultimately fatal disease necessitates ET as potential relief of physiologic and structural alterations and decrease in functional impairment and mortality [27].

Physical activity can be conducted as acute or chronic exercise training, moreover in both performances the exercise type can be of aerobic or anaerobic character. For therapeutic purposes, it can be assumed that different types of ET have various effects on the patient’s status. In the setting of PH, acute training in the form of CPET is mainly used for diagnosis and assessment of severity [33]. On the basis of a CPET, an individualized ET for disease improvement can be established. The most beneficial modalities of ET and rehabilitation have only just been started to be examined in the literature. Haykowsky et al. [41] found a positive effect of aerobic ET on pulmonary vascular and cardiac remodeling as well as left heart failure in comparison to a combined aerobic and anaerobic training. From these findings, it can be deduced that presumably also in PAH the aforementioned key factors and related mechanisms can be positively influenced by such an aerobic ET. On the other hand, Brown et al. [42] confirmed a reversing effect on RV dysfunction and hypertrophy in MCT-induced experimental PAH by conducting high intensity interval training (HIIT), but not in the training mode of continuous aerobic ET. Thus, it is suggested that the exercise modality seems to have an effect on disease-related pathological mechanisms and clinical manifestations. However, randomized controlled human studies are required to validate the superiority of HIIT. In general, ET is often assessed under the growing stream of “exercise as medicine” and was continuously found to have beneficial effect, when the right modality and dose is applied [43]. In healthy individuals as well as in those with chronic diseases, Mitchell and Barlow [43] showed an improvement in a patient’s psychological status, referring to health-related QoL in connection with heart failure, when being physically active. Moderate ET was seen to be beneficial on a long-term perspective but still exercise intolerance there can occur due to predisposition caused by yet hidden comorbidities or by a pathologic right ventricle (as mentioned in Section 2.1). Here, it has to be pointed out that exercise training regardless of its type requires a certain physical status of the patients, which is WHO/NYHA Class I and II rather than Class III and IV. At the early symptomatic phases, the RV is able to handle the demand of increased contractility to ensure sufficient CO because of the increased afterload which is often termed right ventricular pulmonary arterial coupling (“RV-PA coupling”). Due to increasing stiffness of the cardiac tissue and persistent pulmonary vascular afterload, a reduction in RV contractility will be observed leading to a decrease in stroke volume. If this “uncoupling” occurs the RV is not able to maintain this fundamental prerequisite of functional RV capacity which is needed to perform physical training. This demonstrates that “RV-PA coupling” can be a potential cause of exercise intolerance especially for PH patients in an advanced disease stage (i.e., “mal-adaptive” phase) as is the case for WHO/NYHA Class III and IV. This process is thought to be dynamic, as shown by Singh et al. [44] who investigated the effect of maximum incremental exercise on RV-PA coupling in patients with exercise-induced PH and PAH compared to a healthy control group. Here, both patient cohorts, i.e., exercise-induced PH and PAH patients, experienced an “RV-PA uncoupling” as defined by a decrease in the Ees/Ea ratio (Ees, end-systolic elastance, right ventricular contractility; Ea, arterial elastance, right ventricular afterload) during invasive cardiopulmonary exercise testing—mainly driven by the increase in right ventricular afterload (Ea) rather than a reduced right ventricular contractility (Ees). These data further support the performance of exercise testing in clinical diagnosis of PH (but also for other hidden pathologies) as already at the early stages of the disease, RV function is compromised.

Nevertheless, it is important for further disease management—especially for therapists, to provide a well-founded systematic overview of the effects of ET concepts with special emphasis on the type, duration, intensity and setting which impact on the parameters of PAH. As the underlying mechanism for the enormous improvement of functional state and hemodynamics in PAH due to physical activity are largely still unknown, a clinical study (NCT04188756) was recently initiated which aims to evaluate the impact of exercise (acute and chronic) on right ventricular performance in patients suffering from PAH.

## 3. Methods

### 3.1. Search Strategy

A search strategy was established to conduct a systematic literature review (Appendix A). Electronic databases were searched until July 2019, using medical subject heading terms and related terms, as documented in the search protocol in the appendix. The key words mainly used were: “exercise training”, “rehabilitation”, physical activity”, “training” OR“exercise” AND “pulmonary arterial hypertension”, “PAH” OR“pulmonary hypertension”. PubMed, providing research from the Medline database, was used as the predominant source of information for this paper. Furthermore, the Library of the Cochrane Association was consulted for trials, as well as the Web of Science database, to find eligible studies. Limitations were made on English and German language only and the article type was filtered for randomized and non-randomized controlled trials (RCT and non-RCT), observational studies, and clinical trials. Subsequently, the reference lists from prior extracted articles were manually searched to identify additional studies. A deeper understanding on the subject and, thus, an adequate criteria selection for studies to be included was gained after first perusal of identified articles and studies, but also reviews related to the topic (Appendix A).

### 3.2. Eligibility Criteria

First, eligible studies for this review were selected based on full-length articles of RCT and non-RCT, non-controlled trials, observational studies, and case reports. Data articles and brief reports were included, if sufficiently representative and meaningful information was provided. Meeting abstracts and conference papers, as well as protocols for studies were excluded. Both human and animal trials were included, if the essential criterion, an induced PAH through a monocrotaline (MCT) injection in rats or a diagnosed PAH in at least 70% of the patient cohort, was met. An induced PH through hypoxia was not found eligible, since this is defined by another PH classification (Group 3) and, thus, would not be equivalent to PAH. Furthermore, MCT-induced PAH in rats was found reliable to present similar and closely related changes to those in PAH patients [45]. The PAH amongst all human patients had to be diagnosed and defined according to the “Updated Clinical Classification of Pulmonary Hypertension” [17]. In human studies, a minimum age of 18 years was determined eligible, since scientific advances have shown an older mean age at PAH diagnosis in general and differences in the pathology of children and youth compared to adults and the elderly [3]. All studies investigating aerobic (light, moderate, and high intensity) and anaerobic (strength and resistance) exercise interventions for a minimum of two weeks training period were included. A combination of both exercise programs, sometimes complemented with respiratory muscle training (RMT), was also found eligible for inclusion. Conversely, all training programs which were based on coordination, flexibility, and/or exclusive RMT were excluded. Studies comprising a training period less than two weeks or performing only an acute exercise testing were disregarded from this review. Studies not matching all of the above inclusion criteria were found unsuitable and were excluded, as well as by the exclusion criteria, both shown in Table 2.

#### 3.2.1. Inclusion of Studies

After determining the strategy for finding eligible articles and studies, the identification, screening, and decision-making procedure was executed as described in the flow chart in Figure 2. The definition and application of inclusion and exclusion criteria was indispensable to identify reasonable studies for assessment within in this review. However, the search process showed that the existence and availability of adequate studies, examining the effects of chronic exercise training interventions on PAH, is very limited. This field of research is still very topical and, thus, 32 studies included relevant information and were ultimately included to be assessed in this review.

#### 3.2.2. Exclusion of studies

The exclusion of identified full-text articles followed after first perusal and application of the exclusion criteria. Studies not matching the inclusion criteria were excluded due to the reasons referred to in Table 3. Those studies were found inadequate, since comparability and transfer of examined research subjects with the included studies would be restricted.

The study of Woolstenhulme et al. [67] had to be excluded to avoid over-representation of a single population, since the same cohort was analyzed by Chan et al. [69]. Furthermore, the outcome measures did not meet the criteria of this review.

### 3.3. Study Quality Assessment

The various studies included in this review were assessed for their quality using the study quality assessment tool for RCTs and non-RCTs of health care interventions from Downs and Black [70]. This checklist was ranked as one of the top 14 tools for study quality assessment and further holds the advantage of enabling an examination of both RCTs and non-RCTs [71]. The application of this instrument was also found in other systematic reviews with regards to the medical field of PAH, for example in Babu et al. [72]. However, this tool was established to apply to human health care trials and therefore is not applicable and cannot be used to evaluate animal studies. The checklist comprises five categories for evaluation: reporting (10 items); external validity (3 items); internal validity—bias (7 items); internal validity—confounding (6 items); and power (1 item). Each category consists of the aforementioned number of items which are rated by points, adding up to a sore of a total maximum of 32 points. Nevertheless, based on the percentage of the total score reached in the Downs and Black [70] scale, human studies were rated of low (<55%; ≤17), moderate (55–79%; 18–25), and high (≥80%; 26–32) quality. This rating of study quality follows other reviews, which used the same tool, but was adjusted according to more general grading systems with very good, good, and satisfactory classification.

## 4. Results

The previous stated process of literature search, perusal, and exclusion resulted in a total of 32 studies, of which 13 were RCTs, 16 non-randomized clinical trials, and 3 case reports or case series. Twenty of the included studies assessed human subjects and 12 were animal trials with rats. Amongst all human studies, the sample size of PAH patients added up 628 people (male and female). All of the PAH patients in human study cohorts could be classified according to the WHO and/or NYHA functional classification shown in the Appendix A. Looking at animal studies, the number of all examined rats counted 614. All of the trials were conducted, evaluated, and published between 2005 and 2019.

A foundational object in this paper is to evaluate different types of ET and its effects on PAH. Thus, the studies included conducted various modes of exercise interventions in order to examine effects on PAH. Four of the human interventions performed exclusive aerobic ET including running on a treadmill, walking over ground, and cycling. One exclusive anaerobic ET was conducted within a trial of Gerhardt et al. [73], using a side-alternating platform for whole body vibration (WBV) exercises in 22 humans. Moreover, the combination of aerobic exercise and anaerobic exercise training was found in five human studies. This was partially complemented with RMT in further ten studies. Mereles et al. [11] was the first to establish a combinatory ET and to investigate an effect on PH. This particularly established training concept was the basis of seven further studies on PAH. The training mode of aerobic ET was conducted in eleven trials on experimental PAH in rats. These interventions comprised of running on a treadmill, with one modified eccentric treadmill training by Enache et al. [74] and a running wheel offered for voluntary exercising in another animal study by Natali et al. [75]. A high intensity interval training (HIIT) was performed by 42 rats in the study of Brown et al. [42].

The duration of the exercise interventions, whether it was institutional-based, home-based, a combination of both, or an experimental setting in rats, ranged from 2 to 24 weeks. The training setting of an institutional-based ET in combination with a subsequent home-based ET was assessed in eight human studies. However, it should be noted that the studies by Gerhardt et al. [73] and Shoemaker et al. [76] did not contain sufficient information on whether the ET was institutional-based or home-based. All of the patients in institutional-based exercise programs were monitored and supervised by specialist physicians and exercise physiologists. In the home-based exercise settings following the ET of Mereles et al. [11], the patients were advised to closely stay in contact via phone calls with their responsible physicians. Only the trial of Brown et al. [77] purposed to evaluate a mainly unsupervised home-based ET.

The following part of the paper gives an overview of established effects of different ET programs on PAH. Therefore, the results of the objects of investigation were categorized into cellular and molecular, physical, and psychological factors. Table 4 summarizes all of the included human studies, sorted by the type of ET and date and Table 5 summarizes those for included animal studies. Studies were evaluated with regard to the exercise mode and particular emphasis was given to gather information on frequency, intensity, type, and time (FITT), which are the recommended principles for prescribing ET by the ACSM [78]. Further, the examined study outcomes are compared and ET effects on pathological alterations and clinical manifestations are highlighted within the stated subcategories.

Twelve animal studies, as previously stated, were included in the assessment within this review. The experimental design was either RCT (*n* = 5) or non-RCT (*n* = 7), comparing ET and sedentary behavior in experimental PAH and control animals. For the sake of simplicity, the groups are stated as: “MCT + ET” or “MCT + SED” for MCT-induced PAH in rats conducting ET or being sedentary, and “ET” or “sedentary” for rats receiving a saline injection with placebo effect, in Table 5. The experimental PAH in affected rats was induced by an MCT injection in all included animal studies. For animals assigned to control groups, a placebo saline injection was used. The injections (red line in Figure 3) and training programs were scheduled for different time points amongst the studies to be compared, and served the purposes of assessing various effects of ET. In all of the studies [42,45,74,75,79,80,81,82,83,84,85,86], an adaptation period, highlighted in grey in Figure 3, was provided for a time period of 1–5 weeks to familiarize animals with the treadmill running and adjust a similar baseline level. Preventive ET and preconditioning before MCT injections or ET following the PAH onset were examined and described with emphasis on retrieving effects in order to possibly transfer to human PAH etiopathogenesis. The periods of the training interventions are marked in white boxes in Figure 3.
jcm-09-01689-t004_Table 4Table 4Characteristics of included exercise training studies in humans (*n* = 20).**Author (Year)****Design****Cohort****Duration****Exercise Training (FITT)****Aerobic exercise training****[76] Shoemaker et al. (2009)****Pre-post**(case report)**2 humans**male (*n* = 1), age: 50 years, iPAH;female (*n* = 1), age: 57 years, sclerodema**6 weeks****cycling exercise:** 3 days/week; workload: 50% of CPET peak, duration: 35 min**[69] Chan et al. (2013)****RCT****23 females**ET (*n* = 10), age: 53 ± 13 years, WHO I–IV. Control (*n* = 13), age: 55.5 ± 8.5 years, WHO II–III**10 weeks**, outpatient institutional-based**treadmill running:** 24–30 sessions; 70–80% HRR; duration: 30–45 min)**[88] Weinstein et al. (2013)****RCT****24 females**ET (*n* = 10), age: 53.4 ± 12.4 years, WHO I–IV.Control (*n* = 13), age: 55.3 ± 8.7 years, WHO II–III**10 weeks**, outpatient institutional-basedsame as [69] Chan et al. (2013)**[77] Brown et al. (2018)****Pre-post****12 females**age: 44 ± 4 years, NYHA II–III**12 weeks**, home-based**walking exercise:** over ground vs. treadmill; 6 days/week, 65–75% HRR; duration: 25 ↗ 45 min (by end of week 2)**Author (year)****Design****Cohort****Duration****Exercise training (FITT)****Anaerobic exercise training****[73] Gerhardt et al. (2017)****RCT****22 humans**ET (*n* = 11)64% female, age: 65.1 ± 5.0 years, 55% WHO II.Control (*n* = 11), 55% female, 46 ± 3.7 years, 64% WHO II**4 weeks****WBV:** 16 sessions of 1h duration; muscle specific ET**Control-WBV:** no training (Weeks 1–5) and subsequently WBV training (Weeks 5–8)**Author (year)****Design****Cohort****Duration****Exercise Training (FITT)****Combined training interventions–aerobic + strength****[89] Man et al. (2009)****Pre-post****19 humans**79% female,age 42 ± 13 years, WHO II–III**12 weeks**, outpatient institutional-based**interval cycling:** 3 days/week; 50–75% VO_2_ peak; 10–5 sets of 2–5min, rest: 2 min **quadriceps strength training:** 3 days/week; 50–75% 1 RM; 3-times 12–15 repetitions**quadriceps endurance training:** 3 days/week; 30–40% 1 RM; 3–6 sets 30–60 repetitions**[90] Mainguy et al. (2010)****Pre-post**(Case report)**5 humans**80% female, 40 ± 15 years, WHO II–III**12 weeks**, outpatient institutional-based**treadmill running:** 3 days/week; 85% of mean speed at 6 MWD; duration: 15 min**cycling exercise:** 3 days/week; 60% max. workload; duration: 10–15 min**strength training:** 3 days/week; 2-times 10–12 repetitions**[91] Martínez-Quintana et al. (2010)****non-RCT****8 humans**NYHA II–IIIET (*n* = 4), 50% female,age: 22.7 ± 6.9 years; Control (*n* = 4), 25% female, age: 32.7 ± 5.5 years**12 weeks**, outpatient institutional-based**interval cycling:** 2 days/week; 80% HRmax; 24 min of 10–25 W with 0.5 min peaks of 20–50 W**strength training:** 2 days/week; 1–2 kg weights**[92] Fox et al. (2011)****non-RCT****22 humans**91% PAH, NYHA II–III.ET (*n* = 11), 91% femaleage: 57 ± 3.7 yearsControl (*n* = 11), 45% female age: 46 ± 4.5 years**12 weeks**, outpatient institutional-based**block 1 (Weeks 1–6):** interval training 2 days/week; treadmill running, cycling and step climbing; 60–80% HRmax; duration: 60 min**block 2 (Weeks 7–12):****continuous aerobic exercise:** 2 days/week**strength training:** 2 days/week; 0.5–1 kg dumbbells**home-based exercises:** 7 days/week; stair climbing and brisk walking**[53] Ganderton et al. (2011)****RCT****10 humans**WHO II–III.ET (*n* = 5), 100% femaleage: 51 years;Control (*n* = 5), 80% female, age 53 years**12 weeks**, outpatient institutional-based**+ 12 weeks**, home-based**institutional ET:****treadmill running:** 3 days/week; 70% of age predicted HRmax; incline 3%; duration: 10 min**walking:** 3 days/week; 60–70% of age predicted HRmax; duration: 10 min**cycling:** 3 days/week; 70% of age predicted HRmax; duration: 10 min**strength training:** 3 days/week; sit-to-stand, step-up, upper limb; 2x20 repititions, 1–3 kg dumbbells**home ET:****walking:** 3 days/week; HR at 120 bpm; duration: 20 min**strength training:** 3 days/week; sit-to-stand, step-up; HR at 120 bpm; duration: 20 min**Author (year)****Design****Cohort****Duration****Exercise Training (FITT)****Combined training interventions–aerobic + strength + RMT****[11] Mereles et al. (2006)****RCT******30 humans****80% PAH, WHO II–IV; ET (*n* = 15), 67% female,age: 47 ± 12 years; Control (*n* = 15), 67% female, age: 53 ± 14 years**3 weeks**, institutional-based**+ 12 weeks**, home-based**hospital ET:****interval cycling:** 7 days/week; 10–25 min circles of ½ min low and 1 min higher workloads (10–60 W); intensity limited by HRmax**walking:** 5 days/week; flat ground and uphill; duration: 60 min**strength training:** 5 days/week; 0.5–1 kg dumbbells; duration: 30 min**RMT:** 5 days/week, duration: 30 min**home ET:****cycling: 5** days/week; duration: 15–30 min**walking:** 2 days/week**strength training:** every other day; duration: 15–30 min**RMT:** every other day; duration: 15–30 min**[37] Grünig et al. (2011)****Pre-post******58 humans****81% PAH, 72% female, age: 51 ± 12 years, WHO II–IV**3 weeks**, institutional-based**+ 12 weeks**, home-basedsame as [11] Mereles et al. (2006)**[12] Grünig et al. (2012a)****Pre-post******183 humans****73% PAH, 69% femaleage: 53 ± 15 years, WHO I–IV**3 weeks**, institutional-based**+ 12 weeks**, home-basedsame as [11] Mereles et al. (2006)**[93] Grünig et al. (2012b)****Pre-post******21 humans****95% female, age: 53 ± 15 years, WHO II–IV**3 weeks**, institutional-based**+ 12 weeks**, home-basedsame as [11] Mereles et al. (2006)**[36] Becker-Grünig et al. (2013)****Pre-post******20 humans****80% female, age: 48 ± 11 years,WHO II–III**3 weeks**, institutional-based**+ 12 weeks**, home-basedsame as [11] Mereles et al. (2006)**[94] Ley et al. (2013)****RCT******20 humans****80% PAH, WHO II–III;ET (*n* = 20), 80% female, age: 47 ± 8 yearsControl (*n* = 20), 60% female,age: 54 ± 14 years**3 weeks**, institutional-basedsame as the hospital ET by [11] Mereles et al. (2006)**[95] Kabitz et al. (2014)****Pre-post****(case series)******7 humans****57% female, age: 59.6 ± 11.1 years, WHO III–IV**3 weeks**, institutional-based **+ 12 weeks**, home-basedsame as [11] Mereles et al. (2006)**[96] Ehlken et al. (2016)****RCT******87 humans****70% PAH, WHO II–IV;ET (*n* = 46); 57% female, age: 55 ±15 yearsControl (*n* = 41), 51% female, age: 57 ± 15 years**3 weeks**, institutional-based**+ 12 weeks**, home-basedsame as [11] Mereles et al. (2006)**[97] Bussotti et al. (2017)****Pre-post******15 humans****87% female, age: 45 ± 13 years,WHO II–III**4 weeks**, outpatient institutional-based**cycling:** 5 days/week;<70% HRmax; duration: 30 min**strength training:** 5 days/week; 0.5–1 kg; 3-times 10–15 repetitions**RMT:** 5 days/week; duration 10–30 min, slow breathing sessions**[98] González-Saiz et al. (2017)****RCT** (with blocking on sex)****40 humans****90% PAH, NYHA I–III;ET (*n* = 20), 60% female, age: 46 ± 11 yearsControl (*n* = 20), 60% female,age: 45 ± 12 years**8 weeks**, outpatient institutional-based**cycling:** 5 days/week; 50% power output of AT, 20–40 min **strength training:** (3 days/week; circuit training 3 sets, large muscle groups**specific RMT:** 6 days/week twice a day6 MWD, 6-min walking distance; AT, anaerobic threshold; BPM, beats per minute; CPET, cardiopulmonary exercise testing; ET, exercise training; FITT, frequency, intensity, type and time of exercise; HR, heart rate; HRR, heart rate reserve; iPAH, idiopathic pulmonary arterial hypertension; NYHA, New York Heart Association; PAH, pulmonary arterial hypertension; 1RM, one-repetition maximum; RMT, respiratory muscle training; VO_2_, volume of oxygen uptake; WBV, whole-body vibration; WHO, World Health Organization.
jcm-09-01689-t005_Table 5Table 5Characteristics of included animal exercise training studies (*n* = 12).**Author (year)****Design****Cohort****Duration****Exercise Training (FITT)****Aerobic exercise training****[79] Souza-Rabbo et al. (2008)****non-****RCT****32 rats (male)**MCT + ET, MCT + SED, Sedentary**5 weeks****treadmill running**(5/7 days; 0.6–0.9 km/h; 50 min)**[80] Handoko et al. (2009)****RCT****56 rats (male)**stable PH (MCT) + ET, progressive PH (MCT) + ET, stable PH (MCT) + SED, progressive PH (MCT) + SED, ET, Sedentary**4 weeks****treadmill running**(5/7 days; 13.3 m/min; 30 min)**[81] Colombo et al. (2013)****non-****RCT****60 rats (male)**histological analysis (*n* = 24), divided in four groups (*n* = 36):MCT + ET, MCT + SED, ET, Sedentary**3 weeks****treadmill running**(5/7 days; 0.9 km/h equals 60% VO_2_ peak; 60 min)**[82] Colombo et al. (2015)****non-****RCT****30 rats (male)**MCT + ET, MCT + SED, ET, Sedentary**3 weeks**same as [80] Colombo et al. (2013)**[83] Moreira-Gonçalves et al. (2015)****RCT****180 rats (male)**MCT + early ET, MCT + late ET, MCT +S ED, early ET, late ET,sedentary**4 weeks** (early ET) or**2 weeks** (late ET)**treadmill running**(5/7 days; 30 m/min ↘ 25 m/min (in the last week); 60 min)**[84] Colombo et al. (2016)****non-****RCT****32 rats (male)**MCT + ET, MCT + SED, ET, Sedentary**3 weeks**same as [80] Colombo et al. (2013)**[85] Nogueira-Ferreira et al. (2016)****RCT****50 rats (male)**ET (*n* = 25), sedentary (*n* = 25)**4 weeks****treadmill running**(5/7 days; 25 m/min; 60 min)**[86] Pacagnelli et al. (2016)****RCT****32 rats (male)**MCT + ET, MCT + SED, ET, Sedentary**11 weeks****treadmill running**(5/7 days; 0.9 km/h; 45 min)**[74] Enache et al. (2017)****RCT****40 rats (male)**MCT + ET (*n* = 13), MCT + SED (*n* = 13), ET (*n* = 7), sedentary (*n* = 7)**4 weeks****eccentric treadmill running**(5/7 days; 65% VO_2_ peak; 33 min; 15° slope)**[45] Zimmer et al. (2017)****non-****RCT****24 rats (male)**MCT + ET (*n* = 6), MCT + sedentary (*n* = 7), ET (*n* = 5), sedentary (*n* = 6)**3 weeks****treadmill running**(5/7 days; 0.9 km/h; 60 min)**Author (year)****Design****Cohort****Duration****Exercise training (FITT)****High intensity interval training****[42] Brown et al. (2017)****non-****RCT****42 rats (male)**MCT + continuous ET (*n* = 7), MCT + HIIT (*n* = 8), MCT + sedentary (*n* = 10), continuous ET (*n* = 5), HIIT (*n* = 6), sedentary (*n* = 6)**6 weeks****interval treadmill running**(5/7 days; 5 × 5 min cycles of 2 min 85–90% VO_2_R and 3 min 30% VO_2_R)**continuous treadmill running**(5/7 days; 50% VO_2_R; 60 min)**Author (year)****Design****Cohort****Duration****Exercise training (FITT)****Voluntary exercise training****[75] Natali et al. (2015)****non-****RCT****36 rats (male)**MCT + ET (*n* = 6), MCT + sedentary (*n* = 6),ET (*n* = 6), sedentary (*n* = 6), MCT + ½ ET (*n* = 6)**4 weeks****running wheel**(voluntary running)ET, exercise training; FITT, frequency, intensity, type and time of exercise; HIIT, high intensity interval training; HR, heart rate; MCT, monocrotaline; PH, pulmonary hypertension; RCT, randomized controlled trial; SED, sedentary; VO_2_, volume of oxygen uptake; VO_2_R, VO_2_ reserve.


### 4.1. Effects of Exercise Training on Molecular and Cellular Factors

Studies were assessed for effects on distinct molecular and cellular factors, such as brain natriuretic peptide (BNP) and NT-proBNP, and factors related to or impacting the vasodilator NO. Two studies contained results on NO and eleven studies addressed BNP levels. Table 6 gives an overview of identified results.

#### 4.1.1. Brain natriuretic Peptide

The hormone BNP and NT-proBNP levels, which provide information about the degree of cardiac failure in PAH affected subjects, were measured by ten human studies and one animal study; Table 6 shows the results. Gerhardt et al. [73] could not find a change in NT-proBNP levels in a RCT of a four week WBV training. The four-week combined aerobic and strengthening ET with additional RMT of Bussotti et al. [97] did not result in a change of the prognostic cardiac failure factor BNP either. Gonzáles-Saiz et al. [98] assessed the effect of an eight-week combination of aerobic and anaerobic ET with RMT through an RCT, but also found a similar level of NT-proBNP with no time or group effect. In further non-RCT and pre-post measures by Fox et al. [92], Man et al. [89], and Martínez-Quintana [91], neither an increase nor a decrease was stated as well. These studies conducted a 12-week outpatient institutional-based ET combining aerobic interval exercises with moderate to vigorous intensity and complementary strength exercises, two or three days per week. A training period of 15 weeks, with a split of three weeks closely supervised hospital-based ET and 12 weeks of home-based ET, could not produce changes in the blood NT-proBNP levels amongst the patients either [93,96]. The ET comprised of a moderate to vigorous intensity interval cycling, walking, and strengthening exercises, as well as additionally RMT. The same training modality was applied to patients in the study of Becker-Grünig et al. [36]. However, they found a significant increase in NT-proBNP levels (*p* < 0.05) during post training measurements.

In contrast to the above stated findings, Brown et al. [77] demonstrated a decrease in blood BNP levels following a 12-week home-based aerobic ET. Patients performed unsupervised walking exercises at a vigorous intensity for 45 min on six days a week. In alignment with these slightly but not significantly decreased levels in one human cohort, Moreira-Gonçalves et al. [83] found a significant result in animal RCT. It was shown that the BNP mRNA levels significantly decrease (*p* < 0.05) in MCT treated animals, conducting a vigorous four-week aerobic ET immediately after the injection (MCT + early ET), compared to sedentary animals (MCT + SED) and those performing a late two week ET (MCT + lateET).

#### 4.1.2. Nitric Oxide

NO, as an important vasodilator in the endothelium, and its related physiological active molecules, such as endothelial nitric oxide synthase (eNOS) or nitric anion (NO_2_^−^), were evaluated by two studies, as shown in Table 6. In the setting of a moderate intensity aerobic ET for five days a week, each session lasting 60 min, Zimmer et al. [45] found no sufficient increase in lung NO_2_^−^ concentration, eNOS enzyme, or NOS activity, which is a responsible enzyme for the formation of NO in the endothelium. No significant changes could be found amongst the four groups, and Zimmer et al. [45] could not show any significant changes in endothelin-1 receptors (ET-A and ET-B). A stimulation of the ET-B receptor induces the secretion of the important vasodilator NO. The expression of the ET-B receptor was slightly reduced in experimental PAH animals, no differences were shown between sedentary and training groups, but in comparison with their respective control groups. Brown et al. [42] produced similar eNOS results in the MCT-induced rats in the group conducting the same training modality of moderate continuous aerobic exercise. Even though the training period was twice as long (six weeks), no significant changes could be noted. However, in this study, Brown et al. [42] compared continuous aerobic ET with HIIT and detected a significant increase in lung eNOS protein expression in rats injected with MCT and complementary performing HIIT. The training was carried out five times a week with 5 × 5 min cycles of 2 min vigorous and 3 min light treadmill running. The lung eNOS protein expression was significantly higher in MCT + HIIT animals compared to MCT + SED (*p* < 0.05) and sedentary animals (*p* < 0.01).

### 4.2. Effects of Exercise Training on Functional and Physiological Factors

Changes in functional and physiological parameters following ET in the setting of PAH were examined with specific emphasis on retrieving effects on RVH and functional exercise capacity (FEC). RVH was measured by heart autopsy in rats and RV echocardiography in humans. A subject’s functional capacity was measured using the 6 MWD and CPET in order to obtain peak VO_2_ or workload at anaerobic threshold (AT). Table 7a,b contain the summarized study results of 15 studies on RVH and 21 studies on FEC.

#### 4.2.1. Right Ventricular Hypertrophy

Cardiac functional and structural changes in the right heart are mutually dependent and have been evaluated in the setting of ET impacting on PAH by three human studies and twelve animal studies. The results presented in Table 7 are described below. The human RCT by Ehlken et al. [96] and Mereles et al. [11] conducted the same combined ET with additional RMT over a period of 15 weeks in total. Patients were to perform interval cycling every day for three weeks in hospital with additionally five days a week walking, strengthening, and respiratory muscle exercises. The 12-week home ET was comprised of a lower frequency of cycling, walking, strengthening, and RMT. Both studies showed no changes in RVH, evaluated by echocardiography. Neither worse nor better outcome in RVH was detected in the RCT of Gerhardt et al. [73] after the applied four-week WBV intervention either.

Animal studies, unless otherwise specified, were comprised of aerobic exercises performed on a treadmill for five days a week with different durations of program period, intensity, and time of each sessions. Handoko et al. [80] found no changes of RVH in rats with MCT-induced PAH following a four-week moderate aerobic ET for 30 min each session. A similar ET according to the FITT principles was performed in the studies of Colombo et al. [81,82,84] and Zimmer et al. [45]. The rats were running at a moderate intensity for 60 min each session over a period of three weeks. All four trials did not produce changes in RVH measured by RV autopsy comparing baseline and follow up results as well as group effects.

One study found a negative effect in rats with experimental PAH conducting a vigorous eccentric ET for four weeks. The MCT treated group, which performed the ET, significantly (*p* < 0.05) worsened in RVH [74]. In contrast to these findings, Natali et al. [75], who offered voluntary treadmill running for four weeks, could produce slight decrease RVH in MCT treated rats performing ET in comparison to MCT + SED animals. However, the improvement was not statistically significant. Souza-Rabbo et al. [79] showed a statistically significant decrease of RVH in MCT + ET animals compared to MCT + SED rats at Week 3 of a moderate running intervention. There were no results available for comparison at Week 5, as MCT + SED animals died previously, although the duration of the ET was scheduled for that time period. Another study evaluated the effect of a four-week ET before PAH was induced by an MCT injection and could produce significant attenuation (*p* < 0.01) in RVH when comparing MCT + ET with MCT + SED [85]. The training program comprised of aerobic exercise sessions over 60 min with vigorous intensity. Moreira-Gonçalves et al. [83] compared a four-week ET immediately after MCT injection to a two week ET with a delayed onset. It was shown that the RV collagen deposit was reduced in groups conducting ET (MCT + early ET (*p* < 0.001) and MCT + lateET (no statistical significance (SS)) in comparison to sedentary animals with experimental PAH. Furthermore, RVH was significantly reduced in both training groups (*p* < 0.01) with a group and time effect. A difference between the early and late ET could not be detected. Nevertheless, Brown et al. [42] stated a significant disparity between the compared training groups of HIIT versus continuous ET. A decrease of RVH in MCT induced PAH in rats performing HIIT (MCT + HIIT) over six weeks was found in comparison to continuous ET (MCT + ET) and sedentary animals (MCT + SED) (*p* < 0.05). Following a long-term aerobic ET of 11 weeks moderate treadmill running for 45 min each session, Pacagnelli et al. [86] produced a highly significant result of 21% attenuation (*p* = 0.0001) in RVH for experimental PAH animals in the ET group in comparison to likewise injected animals of the sedentary group.

#### 4.2.2. Functional exercise capacity

Various parameters were measured by different exercise tests in order to evaluate the FEC status of PAH patients. The submaximal exercise measures of the 6 MWD were investigated by 20 human studies. CPET, which is considered the gold standard for assessing exercise capacity and intolerance [99], was conducted by 14 studies, including one animal study. All of the results towards changes in FEC following an ET are presented in Table 7.

The previously described exercise intervention established by Mereles et al. [11] running for an overall period of 15 weeks in hospital (3 weeks) and at home (12 weeks) included aerobic anaerobic and respiratory muscle exercises and led to positive outcomes in FEC in all studies utilizing this program [12,36,37,93,95,96]. The distance walked during the 6-min-walking-test increased significantly in all seven studies and ranged from 41 to 111 m with probability values less than 0.003. Maximum VO_2_ was also significantly increased (*p* < 0.05) during CPET following the supervised combined ET, underlining an improved FEC. In contrast to these results, Ganderton et al. [53] produced different outcomes with an increase in 6 MWD (No SS) after the first 12 weeks of supervised institutional aerobic and strengthening exercises, but a decrease of 14 m after another 12 weeks training at home. Even though an enhancement of FEC was assessed during CPET, no SS could be found in increased VO_2_ peak and workload at AT following the 24-week training period. No changes of 6 MWD were shown by Man et al. [89] and Martinéz-Quintana et al. [91], although the first study found an increase in workloads at AT (*p* = 0.003) in the CPET after 12 weeks of interval cycling and quadriceps strengthening. Nevertheless, long-term ET conducted in different training settings for durations of 10–12 weeks stated a prolongation of the 6 MWD by 40 m (*p* = 0.01) in an unsupervised walking regimen by Brown et al. [77], up to 58 m after a combined aerobic and anaerobic ET by Fox et al. [92] and Mainguy et al. [90]. An exclusive treadmill running intervention at a vigorous intensity (70–80% of HRR) conducted for 24–30 sessions over 10 weeks resulted in increased 6 MWD of 53 (*p* = 0.003) and 56 m (*p* = 0.002), but not in improved VO_2_ peak [69,88].

The animal study of Brown et al. [42] compared HIIT with continuous ET and sedentary behavior in experimental PAH for six weeks and showed less decrease in VO_2_ peak for ET groups. The MCT treated groups performing HIIT and continuous ET were significantly less impaired (*p* < 0.01 and *p* < 0.05) in comparison to MCT-induced sedentary rats.

Shoemaker et al. [76] reported about two case studies of moderate cycling intervention for six weeks and found an improvement in parameters of FEC, such as a prolonged 6 MWD by 90 m and 102 m, an increased VO_2_ peak by 4% and 14%, as well as elevated workloads at AT by 46% and 53%. Furthermore, the combination of an aerobic and strengthening ET complemented with RMT performed over four weeks in an outpatient institutional setting significantly improved the patients FEC by an additional 32 m in 6 MWD (*p* < 0.001) and increased VO_2_ peak (*p* < 0.001) [97]. The combinatory ET was also conducted in two RCTs by Gonzáles-Saiz et al. [98] and Ley et al. [94] whereas the training interventions were running for three and eight weeks with moderate sessions every day or every other day. Recorded results indicate an improved FEC as the distance walked for 6 min was enhanced by 91 (*p* = 0.008) and 27 m (*p* = 0.01) and VO_2_ peak significantly improved (*p* < 0.001) as well.

Finally, the training modality in the trial of Gerhardt et al. [73] was a WBV training scheduled for 16 sessions of 1 h within four weeks of intervention. It was found that FEC improved in 6 MWD (39.8 m (*p* = 0.004)) and CPET (VO_2_ peak (*p* < 0.005)) in the early training group compared to the control group, which subsequently received the WBV training.

### 4.3. Effects of Exercise Training on Psychological Factors

The subjectively perceived QoL, which is related to a patient’s disease and health status, was examined by 14 human studies to retrieve effects of ET on this psychological burden. The questionnaires “Short Form 36” (SF-36), “Cambridge Pulmonary Hypertension Outcome Review” (CAMPHOR), “Chronic Respiratory Disease Questionnaire” (CRQ), and “EuroQoL-5D questionnaire” (EQ-5D) were used in the studies as tools for QoL analysis. Thereby, examined changes in QoL are stated in Table 8. The SF-36 and CRQ indicate a better QoL when the outcome score is higher. However, an improved QoL evaluated by the CAMPHOR and LPH is represented by lower scores.

#### Quality of Life

Changes in patients’ psychological status were determined through a shift in their perceived health-related QoL. Following a combined aerobic and anaerobic ET with RMT for three weeks in hospital and twelve weeks workout at home, different results were shown. The hospital training, established by Mereles et al. [11] comprised of an everyday interval cycling of 10–25 min circles of 0.5 min low and 1 min higher workloads. Additionally, walking, strengthening, and respiratory muscle exercises were conducted in hospital. The home-based training included every day aerobic exercises (cycling and walking), strengthening ET and RMT every other day. This training modality was performed in six of the studies examining QoL. Mereles et al. [11] produced an overall improvement in QoL with significant increase in the physical (*p* = 0.013) and mental score (*p* = 0.027). Two other studies confirmed similar results in an improved QoL with a significant time effect after ET (*p* < 0.05) [12,37]. Grünig et al. [93] showed an improvement in QoL with significant increases in the subcategories: physical functioning (*p* = 0.025), general health perception (*p* = 0.049), social functioning (*p* = 0.008), mental health (*p* = 0.033), and vitality (*p* = 0.021). However, Becker-Grünig et al. [36] found a significant improvement in only one score of the SF-36 questionnaire, the bodily pain score (*p* = 0.05). The RCT by Ehlken et al. [96] concluded in non-significant scores of improved QoL, but nevertheless the vitality score produced a significant increase (*p* = 0.036) compared to the control group.

A vigorous aerobic ET combined with strengthening exercises over a period of 2-times 12 weeks, beginning with an outpatient institutional-based and finishing with a home-based intervention in a RCT of Ganderton et al. [53], showed no significant improvements in QoL, neither in the SF-36 nor in the CAMPHOR analysis. On the other hand, Chan et al. [69] found significant effects in both QoL questionnaires (*p* < 0.05) in the ET group in comparison to the control group, following a 10-week vigorous aerobic ET (treadmill running, 24–30 sessions) in the absence of complemented strengthening exercises. Supporting the findings of a vigorous aerobic ET, Brown et al. [77] showed a significantly increased QoL score (*p* = 0.02) in the physical component of SF-36 and additional increases in the subcategories physical functioning (*p* = 0.03) and energy/fatigue (*p* = 0.02). The training modality was an unsupervised 12-week home-based walking regimen. Shoemaker et al. [76] reported two cases of moderate aerobic ET over a six-week period with little increase in QoL, measured by CRQ and CAMPHOR, but could not find significant scores. In contrast to this, Martínez-Quintana et al. [91] showed no changes in the shortened SF-12 questionnaire for QoL after a 12-week training intervention of aerobic and anaerobic exercises in an outpatient institutional-based setting. An eight-week combinatory ET with five days of cycling and three days of strengthening exercises per week produced an improvement in QoL in the mental component of the SF-36 and a statistical significant increase in the physical component (*p* = 0.002) in the ET group [98]. A similar training was applied to patients in the four-week outpatient institutional-based intervention by Bussotti et al. [97]. The combination of moderate cycling and strengthening exercises complemented with an additional RMT for five days every week showed significant decreases in anxiety and depression scores (*p* < 0.01) and simultaneously an increase (*p* < 0.01) in the overall QoL of patients. Finally, Gerhardt et al. [73] found in their study design of a RCT in 22 humans a highly significant improvement in QoL (*p* < 0.001) when comparing the ET group with the control group following a four week WBV training. The highly significant decrease in total LPH score (*p* < 0.001) also reflects an improvement in the patients QoL.

### 4.4. Study Quality

To obtain a comparable baseline of the quality of included human studies, an analysis by means of the Downs and Black [70] checklist was conducted and an appraisal was established. Based on a previous determined percentage of the total score to be achieved, the studies were rated of low (<55%; ≤17), moderate (55–79%; 18–25), and high (≥80%; 26–32) quality. The 20 human studies, presented in Table 9, were sorted by study design and date. Out of eight RCT studies, two were rated as high quality, and the remaining six were rated as moderate quality. However, the moderate quality RCTs still scored toward the upper values. The 12 studies designed as non-RCT, non-controlled trials, and observational studies, varied in their level of quality: five were low, six were moderate, and one was high quality. The appraisal of study quality is of particular importance, as by this rating the comparability of study results and its objectivity, reliability, and validity can be distinguished in a differentiated manner and the transfer of scientific data can be justified. In total, out of the 20 human studies that were included in the review article (Appendix A), 10 did not observe any adverse events, 6 did not address this topic in their publication, and only 4 reported very rarely symptoms of respiratory and/or gastrointestinal infections, (pre-)syncope, back pain, and sore muscles. With only a few exceptions, all participants were treated with adequate medication and could continue to take part in the respective clinical study.

The animal studies do not allow for the application of the assessment items of the tool for human studies, and, thus, for the sake of clarity, a more generalized classification was made.

The twelve included animal studies were overall found to be controlled trials, yet not all of them reported a randomization. Therefore, five studies were assessed as RCTs and can be determined as of higher quality in comparison to non-RCT and observational studies.

## 5. Discussion

The idea of a complementary treatment modality such as exercise interventions in addition to a pharmaceutical therapy has only recently become of interest and was included for PH treatment options [13]. Hence, scientific research still lacks well-founded evidence on beneficial effects of ET and recommendations for clinical practice or outpatient exercise programs.

Within this review, various studies which address changes in PAH factors related to different types of ET were evaluated. However, it has to be clarified that interventions did not only differ in performed types of exercises, but also in terms of the previously stated remaining FITT-principles. It appears that the included studies presented very diverse outcomes regarding effects on causal factors and clinical manifestations.

As previously mentioned, PAH has a multifactorial pathogenesis, which requires extensive investigations to make general statements about the effects of ET on specific factors. The same applies to the broad variety of clinical symptoms and its suggestibility. The factors limited in this paper have been analyzed for exercise-related changes based on the physical activity programs of identified studies. However, not only the clinical picture of PAH, but also the intervention programs of the studies were very heterogeneous and thus made the comparability of the results more difficult. In addition, not all studies took the same target variables into account, thus the causal effects of improved patient statuses were difficult to examine.

Nevertheless, studies showed that chronic physical activity does have an influence on the PAH-related pathophysiological changes, even if it only affected some aspects or demonstrated mixed results. Overall, it is fair to say that the effects of various conducted ETs were mainly positive and cellular, molecular, physiological, and psychological factors were improved, or, if no changes were found, it can be concluded that ET at least does not have a negative impact on PAH patients. Only in one study a deterioration of RVH following aerobic ET could not be ruled out [74]. Another study produced increased levels of NT-proBNP after a combined ET with additional RMT [36]. A third study demonstrated controversial results in terms of patients FEC [53]. Table 10 presents a summarized overview of studies and their effects on discussed factors, whether it was minus (−), neutral (0), or positive (+), in correlation with different types of ET.

The influence of ET on NO, which is an important for vasodilator in endothelial cells of the pulmonary arteries, was examined in only two studies. Moderate endurance training by Zimmer et al. [45] could not produce increased levels of lung NO_2_^−^ or related factors. However, significant increases were found in rats following a HIIT in comparison to moderate aerobic ET, suggesting that HIIT is superior to a moderate exercise training regimen. This presumption can be strengthened with findings by Park and Omi [100], who stated an increase in pulmonary eNOS levels depending on the type of exercise performed. However, the transfer of results to human conditions is generally difficult at this point as it is based only on data from animal studies.

RV function and structure, which could be assessed by analyzing effects on BNP levels and RVH, showed inconsistent results, and effects could not be clearly attributed to specific ET modes. For the prognostic factors of RV and cardiac functioning, NT-proBNP and BNP, it can be stated that ET did not seem to worsen the condition in general. However, it is worth noting that, even the study of Becker-Grünig et al. [36], which produced increased levels of NT-proBNP in the end of the study period, presented an interim decrease of NT-proBNP levels in patients. These positive effects could be justified by a greater commitment of patients in a supervised institutional-based intervention in comparison to the home-based training. A potential connection between supervised exercise interventions and positive effects on PAH factors could be confirmed by Ganderton et al. [53], as in this study outcomes improved during the initial 12 weeks of supervised training, but worsened towards the end of the another 12 weeks of home-based training. On the contrary, Brown et al. [77] was able to produce positive effect through a 12-week home exercise walking regimen. For this aspect, evidence for a final determination in which setting what degree of supervision would be beneficial, is still to be evaluated.

The training status at the beginning of the exercise programs has only been determined in some studies. However, previously conducted exercise training may impact ET-related cellular and physiological factors in PAH patients, as indicated by two studies which reveal several positive effects, such as cardiac remodeling, following a 4–8-week aerobic exercise training and a subsequent MCT injection to trigger PAH in rats [83,86]. Following these results, it is of great importance to make and early diagnosis for PAH in order to begin an individualized ET appropriate for patient’s functional classification [101].

Nevertheless, attention also has to be given to the duration and intensity of training modalities. The controversial effects in terms of reversing RVH or improving RV remodeling should be considered by general knowledge about effects of intensive long-term endurance training, potentially resulting in RV remodeling in healthy athletes [102]. This could be an explanation for often finding neither worsened nor improved RVH following ET. HIIT, with its short-term stress elevation on the body, could bring about an improvement in RVH in direct comparison to an ET with continuous endurance training [42]. These controversial results make it difficult to make well-founded statements about effects and the derivation of training programs for the practice.

The combination of aerobic and anaerobic exercise training partially complemented with RMT is suspected to improve the patient’s FEC the most. As previously stated, the CPET and submaximal 6 MWD presented greatest improvements in patient’s functional capacity. The increase of the 6 MWD exceeded the recommended minimum important distance of 25–33 m in almost all studies [103]. This positive influence of great exceedance in the 6 MWD can be further on assumed to predict much higher long-term outcomes. However, to what extent a long-term effect can be derived for patients conducting combined chronic ET is still uncertain [104].

QoL status could be improved in most of the related studies. Whether the training mode was supervised, home-based, moderate or vigorous intensity did not lead to differences in the assessment. In dependence of performed ET compared to pure counselling without physical activity, the effects in some studies could be validated as effects of ET [11,77].

Finally, we want to point out some limitations. First, very small sizes in study population limit the generalizability of produced outcomes. Second, the applied tool for assessment of study quality by Downs and Black [70] is easy to understand and quickly to execute, but limitations were found on rating baseline comparability and external validity [71] Moreover, due to ethical reasons and blinding of subject’s data valid verification of multiple participation is limited. Consequently, we cannot entirely rule out that same subjects may have participated in more than one study of the same or different investigators. Further, the transfer of the results of the animal studies to human conditions is limited. Idiopathic PAH is not evoked by a known acute lung injury and the clinical course of PAH in humans differs by evolving slowly and progressively over years [105]. Moreover, MCT-induced PAH in rat model cannot replicate human idiopathic PAH as there are differences in the clinical course and pathophysiology such as more evidence of the involvement of the parenchymal lung, more perivascular inflammation and the absence of microangiopathy characterized by plexiform lesions [105]. To date, there is only a very small number of pilot studies investigating the effects of exercise using novel animal models such as the Sugen-Hypoxia (SuHx) rat model [56,106,107] or the pulmonary arterial banding (PAB) mouse model [108]. Due to the small number of exercise studies using the SuHx and/or the PAB model and since direct comparability to the results of MCT-model is limited, we excluded those two models from the analysis. In general, SuHx model seems closer to human pathology due to the occurrence of plexiform lesions in the pulmonary system. However, recent reports indicate that SuHx model in severe PAH does not completely correlate with the amount of occlusive lesions [109,110]. The PAB model is not appropriate to study pulmonary arterial hypertension because the right ventricle is not responding to an increased dynamic load due to higher pulmonary vascular resistance but to pressure overload induced to a fixed constriction of the main pulmonary artery by a surgical procedure. Therefore, we do not recommend replicating existing exercise studies using the SuHx or the PAB model as they do not fully resemble the human pathobiology with respect to the multiple structural changes observed in the cardiopulmonary circuit. Rather, future research investigating the beneficial effects of exercise training should focus more on human-based studies using innovative imaging techniques and cell culture approaches. The long-term goal should be to develop standardized generalized exercise training programs and specific recommendations for patients suffering from PAH in order to increase quality of life and to slow down the disease progression on a physiological and cellular level.

## 6. Conclusions

ET has beneficial effects on the health status and the quality of life of PAH patients who therefore should be encouraged to exhibit physical activity. Individualized instructions and regular clinical monitoring provide optimal benefits and ensure that the minimal risk of adverse events is kept as low as possible. Based on the current study situation, no specific recommendation for the modality, duration, and intensity of exercise training can yet be given. However, it is suggested that vigorous intensities in combination with interval training modes are superior to low-intensity exercise stimulating greater improvements in physiological disease-related factors. A combinatory aerobic and anaerobic training with additional RMT induces the strongest improvement in functional capacity. An early training onset and preconditioning in patients may further improve positive outcomes. The need for effective exercise therapies is present as patients as well as therapists benefit from well-founded ET recommendations to provide therapeutic options and thereby improve or reverse the pathophysiological alterations and clinical symptoms of PAH. The very heterogeneous exercise interventions evaluated within this review require further human-based RCTs investigating standardized exercise protocols to clarify the presented results.

## Figures and Tables

**Figure 1 jcm-09-01689-f001:**
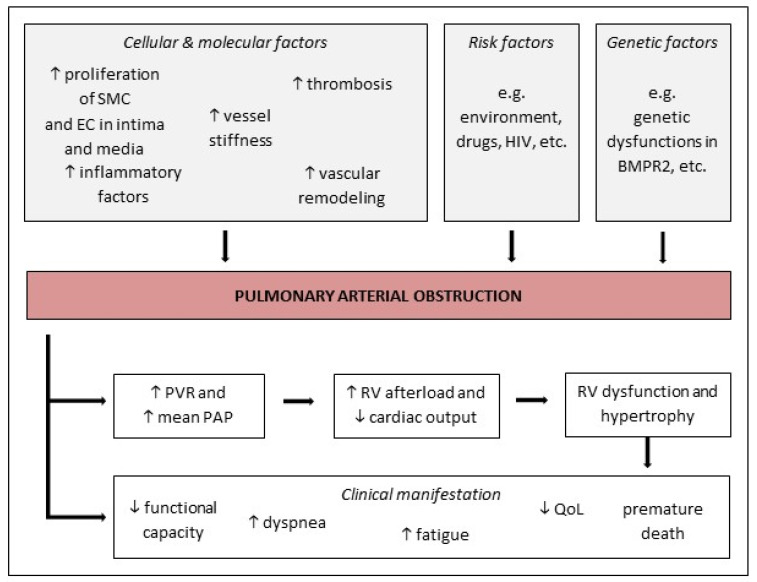
The multifactorial etiopathogenesis of PAH. The pathogenesis of PAH implicates several molecular and cellular factors such as an increased proliferation of SMCs and ECs in the tunica intima and media. Risk factors (e.g., drugs) and genetic factors (e.g., dysfunctions in BMPR2) are also discussed to be involved in pathogenesis. PAH is characterized by an increased PVR and mean PAP ≥ 20 mmHg at rest, progressively leading to right ventricular hypertrophy (RVH), further resulting in right heart failure, loss of functional capacity and premature death. BMPR2, bone morphogenetic protein receptor type II; EC, endothelial cell; HIV, human immunodeficiency viruses, QoL, quality of life; SMC, smooth muscle cells; PAP, pulmonary arterial pressure; PVR, pulmonary vascular resistance (↑ = increased; ↓ = decreased).

**Figure 2 jcm-09-01689-f002:**
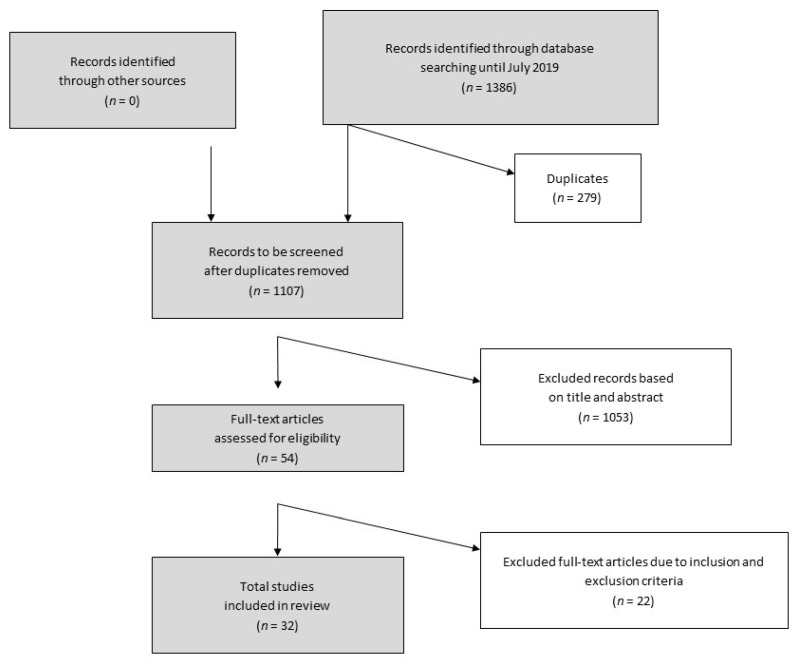
Flow chart of the study inclusion process. A total of 32 studies were finally included to be assessed in this review (modified according to Moher et al. [46]).

**Figure 3 jcm-09-01689-f003:**
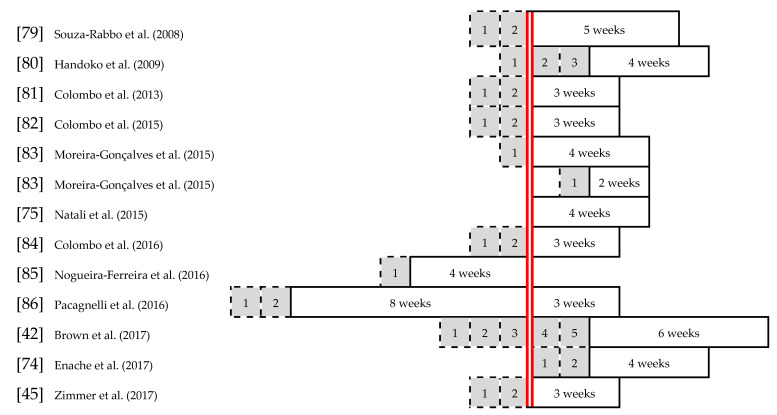
Scheme of exercise interventions in animal studies. The experimental PAH was induced by a monocrotaline (MCT) injection in all included animal studies. The MCT injections (red line) and training programs were scheduled for different time points to assess various effects of exercise training (ET). In all studies [42,45,74,75,79,80,81,82,83,84,85,86], an adaptation period, highlighted in grey, was provided for 1–5 weeks to familiarize the rats with the treadmill running and adjust a similar baseline level. The periods of the training interventions are marked in white boxes (modified according to Moreira-Gonçalves et al. [87]).

**Table 1 jcm-09-01689-t001:** Classification of pulmonary hypertension Group 1.

Group 1—Pulmonary Arterial Hypertension
1.1 Idiopathic PAH
1.2 Heritable PAH
1.3 Drug- and toxin-induced PAH
1.4 PAH associated with:
1.4.1 Connective tissue disease
1.4.2 HIV infection
1.4.3 Portal hypertension
1.4.4 Congenital heart disease
1.4.5 Schistosomiasis
1.5 PAH long-term responders to calcium channel blockers
1.6 PAH with overt features of venous/capillaries (PVOD/PCH) involvement
1.7 Persistent PH of the newborn syndrome

HIV, human immunodeficiency viruses, PAH, pulmonary arterial hypertension; PCH, pulmonary capillary hemangiomatosis; PVOD, pulmonary veno-occlusive disease.

**Table 2 jcm-09-01689-t002:** Eligibility criteria for study inclusion and exclusion.

Inclusion Criteria	Exclusion Criteria
Full-length original article, data and brief report	Meeting abstract, conference paper, study protocol
Human study: diagnosed PAH (PH Group 1)	Human study: PH Group 2–5 (≥30% of the cohort) and healthy population
patients (≥70% of the cohort)	Animal study: hypoxia-induced PH, SuHx
Animal study: MCT-induced PAH	rat model, PAB mouse model
ET: aerobic (light, moderate and high intensity)	ET: coordination, flexibility and exclusive
and anaerobic (strength)	respiratory muscle training
Chronic ET (≥2 weeks)	Acute ET (<2 weeks)
Age ≥ 18 years (human)	Age ≤ 18 years (human)

ET, exercise training; MCT, monocrotaline; PAB, pulmonary artery pressure; PAH, pulmonary arterial hypertension; PH, pulmonary hypertension; SuHx, Sugen-Hypoxia.

**Table 3 jcm-09-01689-t003:** Excluded studies with named reason (*n* = 22).

Author (Year)	Reason for Exclusion
[47] Babu et al. (2019)	cohort → only 40.5% PAH patients
[48] Betancourt Pena et al. et al. (2018)	cohort → PH secondary to lung disease
[49] Brown et al. et al. (2015)	intervention → acute exercise
[50] Chia et al. (2017a)	type of article → study protocol
[51] Favret et al. et al. (2006)	cohort → no MCT-induced PAH in rats
[52] Fowler et al. et al. (2013)	intervention → acute exercise
[53] Ganderton et al. (2011)	type of article → study protocol
[54] Goret et al. et al. (2005)	cohort → no MCT-induced PH in rats
[55] Harbaum et al. (2016)	intervention → acute exercise
[56] Hargett et al. (2015)	cohort → no MCT-induced PAH in rats
[57] Ihle F et al. (2014)	cohort → PH, but no data if ≥ 70% PAH
[58] Laoutaris et al. et al. (2016)	cohort → no PAH
[59] Morris et al. (2018)	type of article → study protocol
[60] Postolache et al. et al. (2018)	cohort → case study; patient with PH, but no PAH
[61] Saglam et al. (2015)	intervention → exclusive inspiratory muscle training
[62] Sanchis-Gomar et al. (2015)	type of article → study rationale and design
[63] Santos-Lozano et al. (2017)	no sufficient/inconsistent information
[64] Spruijt et al. (2015)	intervention → acute exercise
[65] Uchi et al. (2005)	cohort → patients aged 5–37 years
[66] Weissmann et al. (2014)	cohort → no MCT-induced PAH in rats
[67] Woolstenhulme et al. (2019)	outcome measures → examined only LV function
[68] Zöller et al. (2017)	cohort → patients minimum age of 5 years

LV, left ventricular; MCT, monocrotaline; PAH, pulmonary arterial hypertension; PH, pulmonary hypertension.

**Table 6 jcm-09-01689-t006:** Summary of studies examining molecular and cellular factors (*n* = 13).

Author (Year)	Cohort	Exercise Intervention	Duration	Outcome Measures	Results
[89] Man et al. (2009)	human (*n* = 19)	combined training	12 weeks	blood laboratory	= NT-proBNP levels (No SS)
[91] Martínez-Quintana et al. (2010)	human (*n* = 8)	combined training	12 weeks	blood laboratory	= NT-proBNP levels (No SS)
[92] Fox et al. (2011)	human (*n* = 22)	combined training	2 × 6 weeks	blood laboratory	= NT-proBNP levels (No SS)
[93] Grünig et al. (2012b)	human (*n* = 21)	combined training + RMT	15 weeks	blood laboratory	= NT-proBNP levels (No SS)
[36] Becker-Grünig et al. (2013)	human (*n* = 20)	combined training + RMT	15 weeks	blood laboratory	↑ NT-proBNP levels (*p* < 0.05)
[83] Moreira-Gonçalves et al. (2015)	animal (*n* = 180)	early vs. late running	4 vs. 2 weeks	RV sample	↓ BNP mRNA in MCT + early ET vs. MCT + SED and MCT + late ET (*p* < 0.05)
[96] Ehlken et al. (2016)	human (*n* = 87)	combined training + RMT	15 weeks	blood laboratory	= NT-proBNP levels (No SS)
[42] Brown et al. (2017)	animal (*n* = 42)	HIIT vs. continuous running	6 weeks	immuno-blotting	↑ lung eNOS protein expression in MCT + HIIT vs. MCT + SED (*p* < 0.05) and sedentary (*p* < 0.01)
[97] Bussotti et al. (2017)	human (*n* = 15)	combined training + RMT	4 weeks	blood laboratory	= BNP levels (No SS)
[73] Gerhardt et al. (2017)	human (*n* = 22)	WBV	4 weeks	blood laboratory	= NT-proBNP levels (No SS)
[98] González-Saiz et al. (2017)	human (*n* = 40)	combined training + RMT	8 weeks	blood laboratory	= NT-proBNP levels (No SS)
[45] Zimmer et al. (2017)	animal (*n* = 24)	Running	3 weeks	biochemical measures	= lung NO_2_^−^ concentration, NOS activity and eNOS enzyme (No SS)
[77] Brown et al. (2018)	human (*n* = 12)	Walking	12 weeks	blood laboratory	↓ BNP levels (No SS)

eNOS, endothelial nitric oxide synthase; ET, exercise training; HIIT, high intensity interval training; MCT, monocrotaline; NO_2_^−^, nitric anion; NOS, nitric oxide synthase; NT-proBNP, N-terminal pro-brain natriuretic peptide; RMT, respiratory muscle training; RV, right ventricle; SED, sedentary; SS, statistical significance; WBV, whole-body vibration.

**Table 7 jcm-09-01689-t007:** Summary of studies examining functional and physiological factors (*n* = 32). (**a**): Summary of human studies examining functional and physiological factors (*n* = 20). (**b**): Summary of animal studies examining functional and physiological factors (*n* = 12).

**(a)**					
**Autdor (Year)**	**Cohort**	**Exercise Intervention**	**Duration**	**Outcome Measures**	**Results**
**Human studies**					
**[11] Mereles et al. (2006)**	human (*n* = 30)	combined training + RMT	15 weeks	echocardiography 6 MWD CPET	= RVH (No SS), ↑ 6 MWD by 111 m (*p* < 0.001) ↑ VO_2_ peak (*p* < 0.05) ↑ workload at AT (*p* < 0.05)
**[89] Man et al. (2009)**	human (*n* = 19)	combined training	12 weeks	6 MWD CPET	= 6 MWD (No SS) ↑ workload at AT (*p* = 0.003)
**[76] Shoemaker et al. (2009)**	human (*n* = 2)	cycling	6 weeks	6 MWD CPET	↑ 6 MWD by 90 m and 102 m ↑ VO_2_ peak by 4% and 14% ↑ workload at AT 46% and 53%
**[90] Mainguy et al. (2010)**	human (*n* = 5)	combined training	12 weeks	6 MWD	↑ 6 MWD by 58 m (*p* = 0.01)
**[90] Martínez-Quintana et al. (2010)**	human (*n* = 8)	combined training	12 weeks	6 MWD	= 6 MWD (No SS)
**[92] Fox et al. (2011)**	human (*n* = 22)	combined training	2 × 6 weeks	6 MWD CPET	↑ 6 MWD by 58 m (*p* = 0.003) ↑ VO_2_ peak (*p* = 0.02)
**[37] Grünig et al. (2011)**	human (*n* = 58)	combined training + RMT	15 weeks	6 MWD CPET	↑ 6 MWD by 87 m (*p* < 0.001) ↑ VO_2_ peak (*p* < 0.001) ↑ workload at AT (*p* < 0.001)
**[53] Ganderton et al. (2011)**	human (*n* = 10)	combined training	2 × 12 weeks	6 MWD CPET	↑ 6 MWD by 69 m Week 12 (No SS) ↓ 6 MWD by 14 m Week 24 (No SS) ↑ VO_2_ peak (No SS) ↑ workload at AT Week 12 (*p* < 0.032) (No SS at Week 24)
**[12] Grünig et al. (2012a)**	human (*n* = 183)	combined training + RMT	15 weeks	6 MWD CPET	↑ 6 MWD by 68 m at Week 3 and 78 m at Week 15 (*p* < 0.001) ↑ VO_2_ peak (*p* < 0.001)
**[93] Grünig et al. (2012b)**	human (*n* = 21)	combined training + RMT	15 weeks	6 MWD CPET	↑ 6 MWD by 64 m at Week 3 and 71m at Week 15 (*p* < 0.003) ↑ VO_2_ peak (*p* < 0.008)
**[36] Becker-Grünig et al. (2013)**	human (*n* = 20)	combined training + RMT	15 weeks	6 MWD CPET	↑ 6 MWD by 63 m at Week 3 and 67 m at Week 15 (*p* ≤ 0.001) ↑ VO_2_ peak (*p* < 0.01)
**[69] Chan et al. (2013)**	human (*n* = 23)	running	10 weeks	6 MWD CPET	↑ 6 MWD by 56 m (*p* = 0.002) = VO_2_ peak (No SS)
**[94] Ley et al. (2013)**	human (*n* = 20)	combined training + RMT	3 weeks	6 MWD	↑ 6 MWD by 91 m (*p* = 0.008)
**[88] Weinstein et al. (2013)**	human (*n* = 24)	running	10 weeks	6 MWD	↑ 6 MWD by 53 m (*p* = 0.003)
**[95] Kabitz et al. (2014)**	human (*n* = 7)	combined training + RMT	15 weeks	6 MWD	↑ 6 MWD by 92 m at Week 3 and 81m at Week 15 (*p* < 0.001)
**[96] Ehlken et al. (2016)**	human (n = 87)	combined training + RMT	15 weeks	echocardiography 6 MWD CPET	= RVH (No SS) ↑ 6 MWD by 41m (*p* = 0.001) ↑ VO_2_ peak (*p* < 0.001)
**[97] Bussotti et al. (2017)**	human (*n* = 15)	combined training + RMT	4 weeks	6 MWD CPET	↑ 6 MWD by 32 m (*p* < 0.001) ↑ VO_2_ peak (*p* < 0.001)
**[73] Gerhardt et al. (2017)**	human (*n* = 22)	WBV	4 weeks	echocardiography 6 MWD CPET	= RVH (No SS) ↑ 6 MWD by 39.8 m (*p* = 0.004) ↑ VO_2_ peak (*p* < 0.005)
**[98] González-Saiz et al. (2017)**	human (*n* = 40)	combined training + RMT	8 weeks	6 MWD CPET	↑ 6 MWD by 27 m (*p* =0.01) ↑ VO_2_ peak (*p* < 0.001)
**[77] Brown et al. (2018)**	human (*n* = 12)	walking	12 weeks	6 MWD CPET	↑ 6 MWD by 40 m (*p* = 0.01) ↑ VO_2_ peak (*p* = 0.02)
**(b)**					
**Author (year)**	**Cohort**	**Exercise intervention**	**Duration**	**Outcome Measures**	**Results**
		**animal studies**			
**[79] Souza-Rabbo et al. (2008)**	animal (*n* = 32)	running	5 weeks	RV autopsy	↓ RVH at Week 3 in MCT + ET vs. MCT + SED (*p* < 0.05)
**[80] Handoko et al. (2009)**	animal (*n* = 56)	running	4 weeks	RV autopsy	= RVH (No SS)
**[81] Colombo et al. (2013)**	animal (*n* = 60)	running	3 weeks	RV autopsy	= RVH (No SS)
**[82] Colombo et al. (2015)**	animal (*n* = 30)	running	3 weeks	RV autopsy	= RVH (No SS)
**[83] Moreira-Gonçalves et al. (2015)**	animal (*n* = 180)	early vs. late running	4 vs. 2 weeks	RV autopsy	↓ RV collagen deposition in MCT + earlyET (*p* < 0.001) and MCT +late ET (No SS) vs. MCT + SED ↓ RVH in MCT + early ET and MCT + late ET vs. MCT + SED (*p* < 0.01)
**[75] Natali et al. (2015)**	animal (*n* = 36)	voluntary running	4 weeks	RV autopsy	↓ RVH in MCT + ET vs. MCT + SED (No SS)
**[84] Colombo et al. (2016)**	animal (*n* = 32)	running	3 weeks	RV autopsy	= RVH (No SS)
**[85] Nogueira-Ferreira et al. (2016)**	animal (*n* = 50)	running	4 weeks	RV autopsy	↓ RVH in MCT + ET vs. MCT + SED (*p* < 0.01)
**[86] Pacagnelli et al. (2016)**	animal (*n* = 32)	running	11 weeks	RV autopsy	↓ RVH by 21% in MCT + ET vs. MCT + SED (*p* = 0.0001)
**[42] Brown et al. (2017)**	animal (*n* = 42)	HIIT vs. continuous running	6 weeks	RV autopsy CPET	↓ RVH in MCT + HIIT vs. MCT + SED and MCT + ET (*p* < 0.05) less ↓ in VO_2_ peak in MCT + HIIT vs. MCT + SED (*p* < 0.01) and MCT + ET vs. MCT + SED (*p* < 0.05)
**[74] Enache et al. (2017)**	animal (*n* = 40)	eccentric running	4 weeks	RV autopsy	↑ RVH in MCT + ET vs. MCT + SED (*p* < 0.05)
**[45] Zimmer et al. (2017)**	animal (*n* = 24)	running	3 weeks	RV autopsy	= RVH (No SS)

(**a**): 6 MWD, 6-min walking distance; AT, anaerobic threshold; CPET, cardiopulmonary exercise testing; ET, exercise training; HIIT, high intensity interval training; RMT, respiratory muscle training; RV, right ventricle; RVH, right ventricular hypertrophy; SED, sedentary; SS, statistical significance; VO_2_, volume of oxygen uptake; WBV, whole-body vibration. (**b**): 6 MWD, 6-min walking distance; AT, anaerobic threshold; CPET, cardiopulmonary exercise testing; ET, exercise training; HIIT, high intensity interval training; MCT, monocrotaline; RMT, respiratory muscle training; RV, right ventricle; RVH, right ventricular hypertrophy; SED, sedentary; SS, statistical significance; VO_2_, volume of oxygen uptake; WBV, whole-body vibration.

**Table 8 jcm-09-01689-t008:** Summary of studies examining psychological factor–QoL (*n* = 14).

Author (Year)	Cohort	Exercise Intervention	Duration	Outcome Measures	Results
**[11] Mereles et al. (2006)**	human (*n* = 30)	combined training + RMT	15 weeks	SF-36	↑ QoL in physical (*p* = 0.013) and mental (*p* = 0.027) component
**[76] Shoemaker et al. (2009)**	human (*n* = 2)	cycling	6 weeks	CRQ CAMPHOR	↑ QoL in both subjects (no SS) ↑ QoL in one subject (no SS)
**[91] Martínez-Quintana et al. (2010)**	human (*n* = 8)	combined training	12 weeks	SF-12	No improvement in QoL
**[37] Grünig et al. (2011)**	human (*n* = 58)	combined training + RMT	15 weeks	SF-36	↑ QoL (*p* < 0.05)
**[53] Ganderton et al. (2011)**	human (*n* = 19)	combined training	2 × 12 weeks	SF-36 CAMPHOR	No SS in improved QoL No SS in improved QoL
**[12] Grünig et al. (2012a)**	human (*n* = 183)	combined training + RMT	15 weeks	SF-36	↑ QoL (*p* < 0.05)
**[93] Grünig et al. (2012b)**	human (*n* = 21)	combined training + RMT	15 weeks	SF-36	↑ QoL in physical functioning (*p* = 0.025), general health perception (*p* = 0.049), social functioning (*p* = 0.008), mental health (*p* = 0.033) and vitality (*p* = 0.021)
**[36] Becker-Grünig et al. (2013)**	human (*n* = 20)	combined training + RMT	15 weeks	SF-36	No SS in improved QoL ↑ in bodily pain score (*p* = 0.05)
**[69] Chan et al. (2013)**	human (*n* = 23)	running	10 weeks	SF-36 CAMPHOR	↑ QoL in ET (group × time effect) (*p* < 0.05) ↑ QoL in ET (group × time effect) (*p* < 0.05)
**[96] Ehlken et al. (2016)**	human (*n* = 87)	combined training + RMT	15 weeks	SF-36	No SS in improved QoL ↑ in ET group vitality score (*p* = 0.036)
**[97] Bussotti et al. (2017)**	human (*n* = 15)	combined training + RMT	4 weeks	HADS EQ-5D	↓ anxiety and depression (*p* < 0.01) ↑ QoL (*p* < 0.01)
**[73] Gerhardt et al. (2017)**	human (*n* = 22)	WBV	4 weeks	SF-36 LPH	↑ QoL in ET (group × time effect) ↑ QoL (*p* < 0.001) ↓ LPH total score (*p* < 0.001)
**[98] González-Saiz et al. (2017)**	human (*n* = 40)	combined training + RMT	8 weeks	SF-36	↑ QoL in mental component (No SS) ↑ QoL in physical component (time effect) (*p* = 0.002)
**[77] Brown et al. (2018)**	human (*n* = 11)	walking	12 weeks	SF-36	↑ QoL in physical component (*p* = 0.02) ↑ in physical functioning (*p* = 0.03) and energy/fatigue score(*p* = 0.02)

CAMPHOR, Cambridge Pulmonary Hypertension Outcome Review; RQ, Chronic Respiratory Disease Questionnaire; ET, exercise training; LPH, living with pulmonary hypertension questionnaire; QoL, quality of life; RMT, respiratory muscle training; SF-36, Short Form 36; SS, statistical significance; WBV, whole-body vibration.

**Table 9 jcm-09-01689-t009:** Summary of quality assessment in human studies (*n* = 20).

Author (Year)	Reporting	External Validity	Bias	Con-Founding	Power	Total Score	Study Quality
Randomized controlled trials							
[11] Mereles et al. (2006)	10	0	5	5	5	25	moderate
[53] Ganderton et al. (2011)	0	2	6	5	0	23	moderate
[69] Chan et al. (2013)	10	1	5	5	5	26	high
[94] Ley et al. (2013)	10	1	4	4	5	24	moderate
[88] Weinstein et al. (2013)	10	1	4	4	5	24	moderate
[96] Ehlken et al. (2016)	9	1	6	6	5	27	high
[73] Gerhardt et al. (2017)	9	0	4	4	5	22	moderate
[98] González-Saiz et al. (2017)	9	1	6	4	5	25	moderate
Non-randomized controlled trials and observational studies							
[76] Shoemaker et al. (2009)	8	2	3	0	0	13	low
[89] Man et al. (2009)	10	2	5	4	5	26	high
[90] Mainguy et al. (2010)	8	2	2	2	0	14	low
[91] Martínez-Quintana et al. (2010)	6	1	2	0	0	9	low
[92] Fox et al. (2011)	9	2	3	1	5	20	moderate
[37] Grünig et al. (2011)	9	0	4	3	5	21	moderate
[12] Grünig et al. (2012a)	9	0	4	3	5	21	moderate
[93] Grünig et al. (2012b)	9	0	4	3	5	21	moderate
[36] Becker-Grünig et al. (2013)	9	0	4	3	5	21	moderate
[95] Kabitz et al. (2014)	9	0	2	2	0	13	low
[97] Bussotti et al. (2017)	9	1	4	3	0	17	low
[77] Brown et al. (2018)	9	1	5	4	0	19	moderate

**Table 10 jcm-09-01689-t010:** Summary of effects on cellular, molecular, physiological, and psychological factors in correlation with different types of ET (negative (−), neutral (0), or positive (+)).

	BNP	NO	RVH	FEC	QoL	
**aerobic ET**						[79] Souza-Rabbo et al. (2008)
			**+**			[80] Handoko et al. (2009)
			**0**	**+**	**0**	[77] Shoemaker et al. (2009)
				**+**/**0**	**+**	[69] Chan et al. (2013)
						[81] Colombo et al. (2013)
			**0**	**+**		[88] Weinstein et al. (2013)
						[82] Colombo et al. (2015)
	**+**		**0**			[83] Moreira-Gonçalves et al. (2015)
			**+**			[75] Natali et al. (2015)
			**0**			[84] Colombo et al. (2016)
			**0**			[85] Nogueira-Ferreira et al. (2016)
			**+**			[86] Pacagnelli et al. (2016)
			**+**			[74] Enache et al. (2017)
			**-**			[45] Zimmer et al. (2017)
	**0**	**0**	**0**	**+**	**+**	[77] Brown et al. (2018)
**anaerobic ET**	**0**		**0**	**+**	**+**	[73] Gerhardt et al. (2017)
**HIIT**		**+**	**+**	**+**		[42] Brown et al. (2017)
**combined ET**	**0**			**+**/**0**		[89] Man et al. (2009)
				**+**		[90] Mainguy et al. (2010)
				**0**	**0**	[91] Martínez-Quintana et al. (2010)
	**0**			**+**		[92] Fox et al. (2011)
	**0**			**+**/**0**/**−**	**0**	[53] Ganderton et al. (2011)
**combined ET + RMT**				**+**	**+**	[11] Mereles et al. (2006)
			**0**	**+**	**+**	[37] Grünig et al. (2011)
				**+**	**+**	[12] Grünig et al. (2012a)
	**0**			**+**	**+**	[93] Grünig et al. (2012b)
	**-**			**+**	**0**	[36] Becker-Grünig et al. (2013)
				**+**		[94] Ley et al. (2013)
				**+**		[95] Kabitz et al. (2014)
	**0**			**+**	**0**	[96] Ehlken et al. (2016)
	**0**		**0**	**+**	**+**	[97] Bussotti et al. (2017)
	**0**			**+**	**+**	[98] González-Saiz et al. (2017)

The combination +/0 or +/0/− is due to the fact that some studies used different tests. BNP, brain natriuretic peptide; ET, exercise training; FEC, functional exercise capacity; HIIT, high intensity interval training; NO, nitric oxide; QoL, quality of life; RMT, respiratory muscle training; RVH, right ventricular hypertrophy.

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
