# Peer review of "Effects of Different Types of Exercise Training on Pulmonary Arterial Hypertension: A Systematic Review"

_jcm, 2020, doi:10.3390/jcm9061689_

Round 1

Reviewer 1 Report

In this review manuscript, Waller et al. analyzed available literature to find whether different types of physical activity of patients could positively affect the progression of pulmonary arterial hypertension. The authors undertook a systematic search for pre-clinical models and clinical reports. The authors indicate a positive effect of exercise in animal models and patients, resulting in improved quality of life, physiological and phycological components of PAH.

This is a very interesting review paper that can influence further studies in the field.

Minor comments:

  1. Table 1, central alignment of the text in table 1, seems difficult to read. Better table organization is required.
  2. It will be interesting to discuss exercise-induced pulmonary hypertension within the scope of the review.
  3. In Figure 1, some words are cut, so the table is difficult to comprehend. It seems like some information is missing. This is important for the assessment of the manuscript.
  4. Table 2. Again, the central alignment of the text is difficult to read.
  5. In Figure 2, the text appears cut. The inclusion/exclusion process does not require a full-size figure and can be explained in the text
  6. Table 3, please change the central formatting—unclear exclusion criteria.
  7. Table 6. It is unclear what “no SS” means
  8. For Table 7, data on pre-clinical models should be separated from clinical trials. The data need to be better organized. It is hard to comprehend the results from this long table.
  9. In table 10, it is unclear why the study is cited when no results provided, for example, for [53] and [54]. Also, it is confusing to see undefined combinations such as “+/0” or “+/0/-“.

Author Response

Comments by Reviewer #1

  1. Table 1, central alignment of the text in table 1, seems difficult to read. Better table organization is required.

Our response: We thank the reviewer for this comment and edited the table 1 according to this suggestion.

  1. It will be interesting to discuss exercise-induced pulmonary hypertension within the scope of the review.

Our response: We thank the reviewer for this valuable comment and added a respective section to our review. We refer to excellent review articles from very well-recognized experts on the field of PH who describe the principles and findings on this important topic in detail. But as our manuscript focusses on the effects of different types of exercise training on alterations in pulmonary arterial hypertension (e.g. caused by structural remodeling of the cellular lung architecture), we don´t include all the aspects of how the individual health status in general (aside from cellular changes in the pulmonary vasculature) predisposes to the onset of pulmonary hypertension under physical stress conditions. Nevertheless, we tried to address this question to the extent which we feel it surely deserves to be discussed at the respective positions in the text (page 3, line 107-124; page 7, line 253-274, clean version).

  1. In Figure 1, some words are cut, so the table is difficult to comprehend. It seems like some information is missing. This is important for the assessment of the manuscript.

Our response: We thank the reviewer for this comment and edited the figure 1 according to this suggestion.

  1. Table 2. Again, the central alignment of the text is difficult to read.

Our response: We thank the reviewer for this comment and edited the table 2 according to this suggestion.

  1. In Figure 2, the text appears cut. The inclusion/exclusion process does not require a full-size figure and can be explained in the text

Our response: We thank the reviewer for this comment and edited the figure 2 according to this suggestion. We decided to present the inclusion/exclusion criteria for better readability.

  1. Table 3, please change the central formatting—unclear exclusion criteria.

Our response: We thank the reviewer for this comment and edited the formatting of table 3.

  1. Table 6. It is unclear what “no SS” means.

Our response: We thank the reviewer for this comment and edited the figure legend accordingly providing the information that “no SS” stands for “no statistical significance”.

  1. For Table 7, data on pre-clinical models should be separated from clinical trials. The data need to be better organized. It is hard to comprehend the results from this long table.

Our response: We separated data on pre-clinical models from clinical trials and edited the formatting of table 7.

  1. In table 10, it is unclear why the study is cited when no results provided, for example, for [53] and [54]. Also, it is confusing to see undefined combinations such as “+/0” or “+/0/-“.

Our response: We thank the reviewer for this comment and edited the format of the table. Combinations such as “+/0” or “+/0/-“ are now explained in the respective figure legend.

Reviewer 2 Report

Thank you for asking me to review this article. It is an interesting systematic review which attempts to associate exercise modality with outcome in PAH at several levels.  It is well written and comprehensive.  I have only  a few minor comments.

  • The English needs review and editing by a native English speaker
  • The diagrams need revising as text is cut-off in places
  • In Figure 2 it would be useful to have no of studies indicated in the various boxes
  • In Table 3 the rows are misaligned
  • I think the conclusion is understated. “ET has no negative effect on the health status of PAH patients and should not be discouraged.” I think the evidence is that is has benefits and should be encouraged.

Author Response

Comments by Reviewer #2

  1. The English needs review and editing by a native English speaker.

Our response: We thank the reviewer for this advice and have our manuscript checked by a professional English editing service.

  1. The diagrams need revising as text is cut-off in places.

Our response: We thank the reviewer for this comment and arranged the text in all figures and tables accordingly.

  1. In Figure 2 it would be useful to have no of studies indicated in the various boxes.

Our response: We thank the reviewer for this comment and arranged the text in all figures and table accordingly.

  1. In Table 3 the rows are misaligned

Our response: We thank the reviewer for this comment and arranged the text in all figures and table accordingly.

  1. I think the conclusion is understated. “ET has no negative effect on the health status of PAH patients and should not be discouraged.” I think the evidence is that is has benefits and should be encouraged.

Our response: We fully agree with the reviewer with this statement. Our results demonstrate that regular exercise training improves the functional exercise capacity. Further, our results suggest that high-intensity exercise training can partially reverse right ventricular hypertrophy. In terms of human studies, an increase of quality of life was found. Furthermore, we provided a new additional table (Table S3) which shows that the occurrence of severe side effects during exercise training is exceedingly rare (page 28, line 641-646, clean version). Therefore, PAH patients should be encouraged to perform exercise training. We revised the conclusion according to the reviewer' s suggestion and highlighted the benefits of exercise training (page 31, line 767-774, clean version).

Reviewer 3 Report

Waller and colleagues present a review entitled Effects of different types of exercise training on pulmonary arterial hypertension: A systematic review. The manuscript is thoroughly written and comprehensive. The authors take great care to review the pathogenesis of PAH as well as rationale regarding the clinical and molecular data that may be impacted with exercise therapy (though much remains unknown regarding the latter). The data presented regarding the exercise methodology and effects of exercise training on RV function, biomarkers and FEC  are nicely summarized. 

Major Comments:

  1. In the Pathogeneses and Clinical Manifestation section the authors should include a mention of RV-PA uncoupling as a potential pathologic cause of exercise intolerance.  Singh and colleagues published data on PA-RV uncoupling during exercise in PAH patients as well as ePH patients. PMID 31218910. Although this was not an endpoint in any of the trials, it imay be an important consideration as a future clinical trial endpoint for patients undergoing exercise training.
  2. Consider toning down the comment "Thus, the exercise modality seems to have a decisive effect on..." line 223-224. This statement is based upon one study of an animal model. At best it is hypothesis generating, rather than definitive proof that HIIT is superior to chronic ET.
  3. Did the authors employ a method to control for duplication of subject data? There are some articles from the same investigators, presumably with the same participants.  This may lead to an overrepresentation of a single population or overstatement of the importance of an outcome. The authors should comment why they believe there is no duplication of study populations and perhaps list this as a limitation fo the study if this cannot be precisely determined.
  4. In the conclusion, the authors state that "ET has no negative effect on the health status of PAH patients and should not be discouraged". This manuscript mainly focuses on the benefit of exercise training.  Could the authors also show data from each human trial regarding detrimental or adverse events of exercise training? Some such as de Man (65) and Chan (43) sited no adverse effects, however Gerhardt (47) listed some such as back pain, while Grünig (12) reported more serious adverse events. This would provide balanced data that would address the risk of exercise training. Overall the risk is low, but it would be nice to show that from a variety of studies. 
  5. The tables should be presented in landscape view, rather than the current view. The information within the columns appear to be crammed in, landscape view would take up less space and allow for better viewing.

Minor Comments:

  1. Words in Figure 1 and 2 need to be adjusted.  Many of the words are cut off and the spaces between lines are too large and need to be adjusted.
  2. For Figure 2, please define "Records identified through other sources" in the Figure legend.
  3. It would be helpful to list the number of records per stage/box in Figure 2. 
  4. Table 3. The manuscripts do not line up with the reason for exclusion. It appears that the shift occurred at Reference #87 on the left and the second to last exclusion rationale on the right.
  5. Consider rewording the sentences on line 294-297, "However, it should..." The wording is somewhat confusing and can be stated more clearly.
  6. Table 4. There are inconsistencies in terminology, female should be spelled out rather than writing "fem" for reference 43.
  7. Each table should have a key for the abbreviations for example Table 4: Please define RMT, 1RM, VO2R etc...
  8. Perhaps list the breakdown of the NYHA or WHO Functional Class of each human study listed in a table.
  9. Table 7 should be split by human and veterinary studies.  I don't think it works having them mixed within the table. 
  10. In Table 9 the second panel of study names do not line up with scores.
  11. Line 657 should be adjusted from "importance to make an early diagnose for PAH" to "importance to make and early diagnosis of PAH".
  12. Reference 65 should be "de Man" rather than "Man".
  13. There are inconsistencies throughout the manuscript with reporting p values, some with a 0 preceding the decimal and others without.  This needs to be adjusted.

Author Response

Comments by Reviewer #3

Major comments:

  1. In the Pathogeneses and Clinical Manifestation section the authors should include a mention of RV-PA uncoupling as a potential pathologic cause of exercise intolerance. Singh and colleagues published data on PA-RV uncoupling during exercise in PAH patients as well as ePH patients. PMID 31218910. Although this was not an endpoint in any of the trials, it may be an important consideration as a future clinical trial endpoint for patients undergoing exercise training.

Our response: We thank the reviewer for this suggestion. We introduce the terminology of “coupling” and “uncoupling” in the Pathogeneses and Clinical Manifestation section (page 4, line 164-169, clean version) and discuss the given citation at an adequate position (page 7, line 265-274, clean version).

  1. Consider toning down the comment "Thus, the exercise modality seems to have a decisive effect on..." line 223-224. This statement is based upon one study of an animal model. At best it is hypothesis generating, rather than definitive proof that HIIT is superior to chronic ET.

Our response: The reviewer is right that it should be noted that the transferability of studies using an animal model to human conditions is limited. In the revised version, we point out that the study only suggests that HIIT is superior to chronic ET and that the results must be validated in human studies (page 6, line 246-248, clean version).

  1. Did the authors employ a method to control for duplication of subject data? There are some articles from the same investigators, presumably with the same participants. This may lead to an overrepresentation of a single population or overstatement of the importance of an outcome. The authors should comment why they believe there is no duplication of study populations and perhaps list this as a limitation for the study if this cannot be precisely determined.

Our response: We thank the reviewer for this valuable comment. If the investigators explicitly refer to the use of the same group of volunteers, we have only included one study. However, we must admit that we did not employ a specific method to control duplication of subject data. Due to ethical reasons and blinding of subject’s data valid verification of multiple participation is limited. Consequently, we cannot entirely rule out that same subjects may have participated in more than one study of the same or different investigators. Therefore, we acknowledged this limitation in our manuscript (page 31, lines 738-741, clean version).

  1. In the conclusion, the authors state that "ET has no negative effect on the health status of PAH patients and should not be discouraged". This manuscript mainly focuses on the benefit of exercise training. Could the authors also show data from each human trial regarding detrimental or adverse events of exercise training? Some such as de Man et. al and Chan et al. cited no adverse effects, however Gerhardt et al. listed some such as back pain, while Grünig et al. reported more serious adverse events. This would provide balanced data that would address the risk of exercise training. Overall the risk is low, but it would be nice to show that from a variety of studies.

Our response: We appreciate this thoughtful comment and edited the manuscript accordingly by implementing a new table (Table S3) which summarizes the authors´ statements about the occurrences of severe side effects (page 28, line 641-646; page 32, line 768-770, clean version).

  1. The tables should be presented in landscape view, rather than the current view. The information within the columns appear to be crammed in, landscape view would take up less space and allow for better viewing.

Our response: We appreciate the reviewer's recommendation to adjust the formatting of the tables. We edited the formatting of the large tables 4, 5, 7 and 8 for better clarity.

Minor comments:

  1. Words in Figure 1 and 2 need to be adjusted. Many of the words are cut off and the spaces between lines are too large and need to be adjusted.

Our response: We thank the reviewer for this comment and edited the manuscript accordingly.

  1. For Figure 2, please define "Records identified through other sources" in the Figure legend.

Our response: We thank the reviewer for this comment and edited the manuscript accordingly.

  1. It would be helpful to list the number of records per stage/box in Figure 2.

Our response: We thank the reviewer for this comment and edited the manuscript accordingly.

  1. Table 3. The manuscripts do not line up with the reason for exclusion. It appears that the shift occurred at Reference #87 on the left and the second to last exclusion rationale on the right.

Our response: We thank the reviewer for this comment and edited the manuscript accordingly.

  1. Consider rewording the sentences on line 294-297, "However, it should..." The wording is somewhat confusing and can be stated more clearly.

Our response: We removed the sentence from the manuscript because it contains redundant information. Though, we stated more precisely that we excluded the study of Woolstenhulme et al.in order to avoid duplication and overrepresentation of a single study population amongst other reasons (page 10, lines 342-344, clean version).

  1. Table 4. There are inconsistencies in terminology, female should be spelled out rather than writing "fem" for reference 43.

Our response: We thank the reviewer for this comment and edited the manuscript accordingly.

  1. Each table should have a key for the abbreviations for example Table 4: Please define RMT, 1RM, VO2R etc...

Our response: We thank the reviewer for this advice and have added the key for the abbreviations under each table.

  1. Perhaps list the breakdown of the NYHA or WHO Functional Class of each human study listed in a table.

Our response: We listed the breakdown of the NYHA or WHO Functional Class of each human study in a supplementary table (Table S4).

  1. Table 7 should be split by human and veterinary studies. I don't think it works having them mixed within the table.

Our response: We thank the reviewer for this suggestion and edited the manuscript accordingly.

  1. In Table 9 the second panel of study names do not line up with scores.

Our response: We thank the reviewer for this comment and edited the manuscript accordingly.

  1. Line 657 should be adjusted from "importance to make an early diagnose for PAH" to "importance to make and early diagnosis of PAH".

Our response: We have revised the wording according to the reviewer's suggestion (page 30, line 712-714, clean version).

  1. Reference 65 should be "de Man" rather than "Man".

Our response: We thank the reviewer for this comment and edited the manuscript accordingly.

  1. There are inconsistencies throughout the manuscript with reporting p values, some with a 0 preceding the decimal and others without. This needs to be adjusted.

Our response: We thank the reviewer for this comment and edited the manuscript accordingly.